# A Geometric Lens on Physics-Aligned Data Compression

Aleix Seguí [1]    Wesley Armour [1]

## Abstract

In AI for Science, physics-informed losses are increasingly used to train learned compressors for scientific data, but their rate–distortion implications remain poorly understood. At fixed bitrate, these objectives often improve preservation of a target physical observable while degrading standard reconstruction fidelity. We develop a local geometric theory showing that this tradeoff is governed by the interaction of latent-space sensitivities induced by the entropy model, the physical observable, and the distortion metric. At each operating point, these induce preferred directions along which compression noise should be suppressed, yielding an anisotropic error-allocation mechanism. When these directions are misaligned, improving the observable at fixed rate necessarily worsens standard distortion, establishing a fundamental limit on simultaneous preservation. We formalise this through a local tangent-space rate–distortion law and introduce a practical alignment diagnostic based on dominant eigenspace overlap. Experiments across scientific domains test the theory and validate that the alignment diagnostic correlates with observed data- and physics-space trade-offs.

## 1. Introduction

Scientific instruments and large-scale simulations in climate, fluid dynamics, geophysics, and astrophysics now produce data volumes that make storage and transfer a primary bottleneck. Yet scientific use rarely depends on pointwise fidelity alone: conclusions are drawn from derived *physical observables* (spectra, gradients, fluxes, conserved quantities, or event indicators) whose sensitivity to perturbations is structured and often highly anisotropic.

[1]Department of Engineering Science, University of Oxford, Oxford, UK. Correspondence to: Aleix Seguí <aleix.seguiugalde@eng.ox.ac.uk>, Wesley Armour <wes.armour@oerc.ox.ac.uk>.

*Proceedings of the $43^{rd}$ International Conference on Machine Learning*, Seoul, South Korea. PMLR 306, 2026. Copyright 2026 by the author(s).

Learned compression offers strong rate–distortion performance through latent representations and learned entropy models (Ballé et al., 2018; Minnen et al., 2018). This flexibility has motivated physics-aware objectives that aim to preserve downstream observables rather than mean-square error (MSE) alone. In scientific compression, this trend appears both in classical quantities-of-interest-aware pipelines and in more recent learned approaches (Ainsworth et al., 2019; Jiao et al., 2022; Lee et al., 2022; Liu et al., 2021a; Galletti et al., 2026). Empirically, such objectives reveal a recurring phenomenon: at fixed bitrate, one can improve preservation of the target observable, but typically at the cost of worse standard reconstruction fidelity. However, the phenomenon remains poorly understood:

*When can a codec preserve physically meaningful quantities under a bitrate constraint, and what governs the resulting trade-offs with standard fidelity measures?*

We address these questions through a local geometric theory of physics-aware compression. Our main result is that, at each latent operating point, preserving the physical observable and preserving standard fidelity each prioritise certain eigendirections along which compression noise should be suppressed. The key feature is whether these directions are aligned. When they are not, improving physical observables at fixed rate necessarily worsens standard distortion, providing a geometric explanation for the fixed-rate physics–MSE tradeoff often seen in practice.

Formally, we show that, under local perturbations in latent space, physics-aware compression behaves as a sample-conditioned tangent-space rate–distortion law. In this regime, error allocation is governed by three latent-space metrics: a *rate* metric induced by the entropy model, a *physics sensitivity* metric induced by the observable of interest, and a *signal fidelity* metric induced by signal-space distortion. This yields an explicit inverse-stiffness allocation rule for compression noise. Our theory further motivates a practical alignment measure based on dominant eigenspace overlap.

We validate these predictions across several regimes, including computational fluid dynamics, cosmological simulations, and electron microscopy measurements of cerebral cortex volume.

## 2. Related Work

**Foundations and learned compression.**  Our work builds on the classical rate–distortion tradition (Shannon, 1948; 1959), high-rate quantisation theory (Gersho, 1979), and modern learned compression with autoencoders and learned entropy models (Ballé et al., 2018; Minnen et al., 2018). Recent advances include expressive entropy and context models, including transformer-based priors (Qian et al., 2022), joint global–local hierarchies (Kim et al., 2022), and data-driven dictionary priors (Lu et al., 2025). Related progress on rate–distortion of discrete representations also appears on task-based vector quantisation (van den Oord et al., 2018; Shlezinger et al., 2019; Gao & Long, 2024; Zandieh et al., 2025). Our contribution differs in focusing on the local geometry induced by the entropy model and its interaction with physics-aware distortion criteria.

**Scientific compression.**  Scientific data compression has a history in high-performance computing, with widely used compressors prioritising throughput and explicit error guarantees on floating-point data (Lindstrom, 2014; Di & Cappello, 2016; Ballester-Ripoll et al., 2020; Di et al., 2025). More recently, both classical and learned approaches have increasingly emphasised preservation of downstream quantities of interest (QoI) rather than only primary-data fidelity. Some works derive explicit theory linking primary-data error to QoI error (Jiao et al., 2022), while others combine learned and classical components to better preserve derived quantities under compression constraints (Ainsworth et al., 2019; Lee et al., 2022; Banerjee et al., 2022; Liu et al., 2021a; 2023; Jia et al., 2025). Domain-specific learned compressors further incorporate physics-informed objectives for particular PDE systems, including turbulent flows and plasma simulations (Momenifar et al., 2022; Choi et al., 2021; Galletti et al., 2026). Our work is complementary: rather than proposing another physics-aware objective, we explain geometrically when such objectives induce a trade-off with standard fidelity and when they can be rate-efficient.

**Indirect, semantic, and task-defined distortion.**  Our formulation is adjacent to information-theoretic treatments where distortion is imposed on an indirectly observed or task-relevant quantity. Early examples include noisy source coding formulations (Dobrushin & Tsybakov, 1962; Wolf & Ziv, 1970) and indirect rate–distortion (Witsenhausen, 1980). Task-aware compression is also closely connected to the information bottleneck (Tishby et al., 2000; Alemi et al., 2019), semantic rate–distortion (Liu et al., 2021b), task-based learned codecs in vision (Duan et al., 2020), and perception (Blau & Michaeli, 2019). Our setting is related in spirit, but differs in developing a sample-local tangent-space characterisation, rather than a global operational rate–distortion function.

## 3. Preliminaries

### 3.1. Learned compression setting

We consider a standard learned compression architecture consisting of an encoder–decoder pair and a learned entropy model. Given data $x \in \mathcal{X} \subset \mathbb{R}^d$, the encoder $f_\phi$ maps the input to a latent representation $z = f_\phi(x) \in \mathcal{Z} \subset \mathbb{R}^m$, which is quantised to $\hat{z}$ and decoded as $\hat{x} = g_\theta(\hat{z})$. The latent representation $\hat{z}$ is modelled via a learned probability density $p_\psi(\hat{z})$.

### 3.2. Physical observables

We assume that data $x \in \mathcal{X}$ arise as observable realisations of an underlying physical system, and that the scientifically meaningful content of the data can be summarised through suitably chosen *observables*. These observables express prior knowledge about which aspects of the signal should be preserved by compression.

**Definition 3.1** (Physical Observable). Let $Q : \mathcal{X} \to \mathcal{Q}$ be a differentiable map from data space to a space of physical quantities of interest (QoIs), such as spectra, gradients, fluxes, conservation laws or any other physical quantities. We assess fidelity with respect to physical semantics through a smooth distortion loss

$$D_Q : \mathcal{Q} \times \mathcal{Q} \to \mathbb{R}_+, \qquad (x, \hat{x}) \mapsto D_Q(Q(x), Q(\hat{x})),$$

which measures deviation in observable space.

The choice of $Q$ specifies which features of $x$ are deemed physically relevant, while $D_Q$ determines how discrepancies in those features are penalised. We introduce a running example of physical observable on Fig. 1.

### 3.3. Problem formulation

Physics-aligned compression can be viewed as an *indirect* rate–distortion problem: rather than constraining reconstruction error only in signal space, we also constrain distortion in the space of physically meaningful observables. The aim is to minimise the required coding rate while ensuring (i) accurate preservation of the observables $Q(X)$ and (ii) acceptable overall reconstruction fidelity.

**Definition 3.2** (Physics-Aware Rate–Distortion). Let $X$ be a random variable on $\mathcal{X}$, and let $Q$ and $D_Q$ be as in Definition 3.1. Let $D_X : \mathcal{X} \times \mathcal{X} \to \mathbb{R}_+$ be a data-space distortion (e.g. mean-square error). For thresholds $(\varepsilon_Q, \varepsilon_X)$, define the physics-aligned rate-distortion function

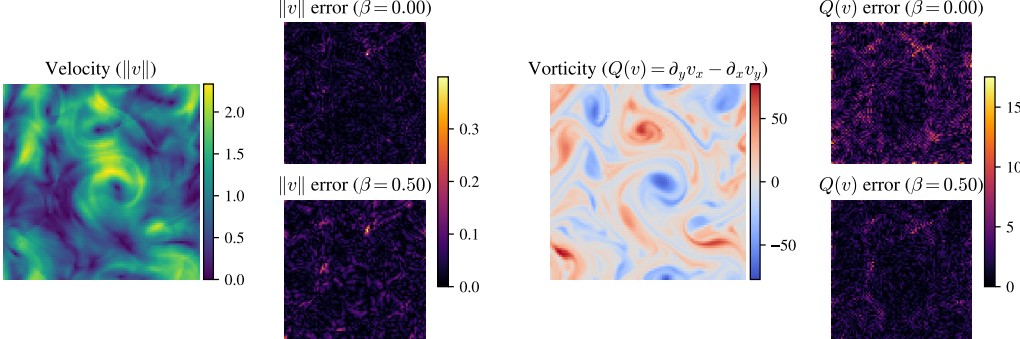

*Figure 1.* As a running example, we use 2D velocity fields from PDEBench (Takamoto et al., 2022) with channels $(v_x, v_y)$. The physical observable is vorticity, $Q(\mathbf{v}) = \partial_y v_x - \partial_x v_y$. The figure compares pointwise compression errors for the reconstructed velocity field and for the derived vorticity. Two models are trained at the same bitrate, 0.85 bps: $\beta = 0$ corresponds to MSE-only training, while $\beta = 0.5$ includes the physics loss, where $\beta$ is the physics weight defined in Section 6. At the same bitrate, the $\beta = 0.5$ model changes the relative allocation of error between velocity-space and vorticity-space distortions.

$$R(\varepsilon_Q, \varepsilon_X) := \inf_{p(\hat{X}|X)} I(X; \hat{X}) \qquad (1)$$

$$\text{s.t.} \quad \mathbb{E}\Big[D_Q\Big(Q(X), Q(\hat{X})\Big)\Big] \leq \varepsilon_Q, \quad (2)$$

$$\mathbb{E}\Big[D_X\Big(X, \hat{X}\Big)\Big] \leq \varepsilon_X. \qquad (3)$$

The observable constraint (2) ensures that compression preserves the prescribed physical quantities of interest, while the signal-space constraint (3) prevents degenerate solutions that may satisfy (2) yet exhibit severe artifacts or implausible deviations along directions weakly constrained by the chosen observable. This formulation is closely related to indirect and semantic rate–distortion viewpoints (Dobrushin & Tsybakov, 1962; Liu et al., 2021b), where the semantics are accessed through $Q(X)$.

# 4. A Local Geometric Model of Physics-Aligned Compression

We now develop a local geometric model that characterises how rate constraints and physics-aware distortion jointly shape admissible reconstruction error. The key idea is to analyse compression through small perturbations in latent space, under which both rate and distortion admit tractable second-order structure.

## 4.1. Local perturbation model in latent space

We adopt a local viewpoint in which the effect of quantisation and entropy coding is modelled as a small random perturbation of a reference latent representation. Let $Z = f_\phi(X)$ denote the latent induced by the encoder, and let $\hat{Z}$ denote the quantised latent. Locally around a typical

$z = f_\phi(x)$, we model

$$\hat{Z} = Z + \eta, \quad \mathbb{E}[\eta \mid X] = 0, \quad \mathbb{E}[\eta\eta^\top \mid X] = \Sigma, \quad (4)$$

where $\eta$ captures compression-induced uncertainty (e.g. quantisation noise) and $\Sigma \succ 0$ is its covariance. Throughout, we interpret $\Sigma$ as the object through which the codec allocates reconstruction error across latent directions.

When needed, we will use the Gaussian local channel approximation $q_\Sigma(\hat{z} \mid x) \approx \mathcal{N}\big(z(x), \Sigma\big)$, which is both analytically convenient and (among all noise models with covariance $\Sigma$) maximises conditional entropy, yielding a conservative "volume" proxy (Gersho & Gray, 1992).

## 4.2. Entropy curvature and a local rate surrogate

To solve the physics-aligned rate-distortion problem defined in Eq. (1), we must quantify the cost of storing the quantised latent $\hat{Z}$. However, directly minimising the mutual information $I(X; \hat{X})$ is usually intractable.

Instead, learned codecs use an entropy model $p_\psi(\hat{z})$ and optimise a variational latent-rate proxy. If $q(\hat{z} \mid x)$ denotes the conditional distribution over quantised latents induced by the encoder and noise model, we write

$$R_{\text{var}} := \mathbb{E}_{p(x)}\Big[\text{KL}\Big(q(\hat{Z} \mid X) \,\|\, p_\psi(\hat{Z})\Big)\Big]. \qquad (5)$$

This is a standard variational upper bound used in learned compression (Ballé et al., 2018), which bounds the rate by the expected KL divergence between the posterior and the prior. Indeed, $I(X; \hat{X}) \leq I(X; \hat{Z}) \leq R_{\text{var}}$ as shown in Appendix A.1.

We now examine this rate locally around a fixed sample $x$, with latent $z = f_\phi(x)$, under the perturbation model (4).

**Proposition 4.1** (Local expansion of the variational rate). *Assume the local Gaussian channel approximation $q_\Sigma(\hat{z} \mid$*

$x) \approx \mathcal{N}(z, \Sigma)$ *with* $\Sigma \succ 0$, *and assume that the negative log-likelihood admits a second-order expansion around $z$ of the form*

$$-\log p_\psi(z+\eta) = \text{const.} + \tfrac{1}{2}\eta^\top H_R(z)\,\eta + o(\|\eta\|^2), \quad (6)$$

*for some symmetric matrix $H_R(z) \succeq 0$. Then the $\Sigma$-dependent part of $R_{\text{var}}$ satisfies*

$$R_{\text{var}} = \text{const.} + \frac{1}{2}\,\text{Tr}\left(H_R(z)\,\Sigma\right) - \frac{1}{2}\log\det\Sigma + o(\|\Sigma\|). \tag{7}$$

The two terms in (7) have different origins. The negative entropy of the Gaussian local channel contributes $-\frac{1}{2}\log\det\Sigma$, penalising shrinking the total volume. The expected negative log-likelihood contributes $\frac{1}{2}\text{Tr}(H_R(z)\Sigma)$, penalising placing uncertainty in latent directions with high entropy-model curvature. The full derivation is given in Appendix A.2.

We therefore use the following local rate surrogate:

$$R_{\text{sur}}(\Sigma; z) := \frac{1}{2}\underbrace{\text{Tr}\left(H_R(z)\Sigma\right)}_{\text{variance placement}} - \frac{1}{2}\underbrace{\log\det\Sigma}_{\text{uncertainty volume}}, \tag{8}$$

dropping additive constants independent of $\Sigma$. This surrogate makes the geometry of rate explicit: at fixed uncertainty volume, noise aligned with high-curvature directions of the entropy model costs more bits than noise aligned with low-curvature directions, as illustrated in Fig. 2.

In practice, $H_R$ should be understood as an effective positive-semidefinite rate metric rather than necessarily the raw Hessian of an arbitrary marginal density. For conditional entropy models such as hyperpriors or context models (Minnen et al., 2018), standard conditional likelihoods often give a PSD curvature in $z$ for fixed side information. When the exact Hessian is indefinite or inconvenient, one can instead use a PSD surrogate, such as a Fisher-information approximation. The subsequent theory only requires such a PSD local metric satisfying the expansion in (7).

### 4.3. Signal fidelity and physical sensitivity

We now quantify how latent perturbations translate into errors both in the reconstructed field and the physical quantities of interest. Under a local tangent-space expansion, both signal-space and physics-aware distortions become quadratic forms in the latent perturbation $\eta$. Their expected values are therefore controlled by the covariance $\Sigma$ introduced in (4).

We fix an operating point $x$ with latent code $z = f_\phi(x)$, and consider a perturbation $\hat{z} = z + \eta$. Linearising the decoder around $z$ gives

$$\hat{x} = g_\theta(\hat{z}) \approx g_\theta(z) + J_g(z)\eta, \tag{9}$$

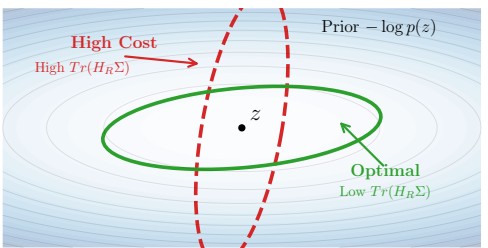

*Figure 2.* **The Geometry of Rate.** The contours represent the negative log-prior $-\log p(z)$. The curvature $H_R$ is high (steep) along the vertical axis and low (flat) along the horizontal. Both ellipses represent a noise covariance $\Sigma$ with the same quantisation volume (same entropy/$\log\det\Sigma$). The red dashed ellipse pays a high bit-cost because it has high variance along the steep direction. The green solid ellipse is optimal: it aligns its variance with the flat directions of the prior, minimising the trace cost $\text{Tr}(H_R\Sigma)$.

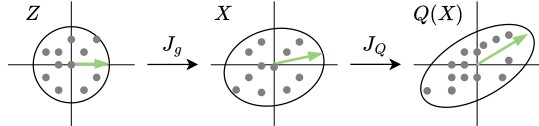

*Figure 3.* Illustration of the error mapping from latent space to physical observable space through $J_Q(x)\,J_g(z)\,\eta$.

where $J_g(z) := \nabla_z g_\theta(z)$ is the decoder Jacobian. If deterministic reconstruction bias at the operating point is neglected, or treated separately, we may write $\delta x := \hat{x} - x \approx J_g(z)\eta$. Passing this perturbation through the observable map yields

$$Q(\hat{x}) \approx Q(x) + J_Q(x)\,\delta x \approx Q(x) + J_Q(x)\,J_g(z)\,\eta, \tag{10}$$

where $J_Q(x) := \nabla_x Q(x)$. Thus, to first order, latent perturbations affect the observable through the composed linear operator $J_Q(x)J_g(z)$, as illustrated in Fig. 3.

To relate these perturbations to distortion, we approximate the losses by local quadratic forms. Let $H_D(x) \succeq 0$ denote the Hessian of the data-space loss with respect to perturbations in $x$, and let $H_\ell(x) \succeq 0$ denote the Hessian of the observable-space loss with respect to perturbations in $Q(x)$.[1] Pulling these quadratic forms back to latent space gives the following two metrics.

**Definition 4.2** (Fidelity and Sensitivity Metrics). The *Fidelity Metric* $G_{\text{eff}}$ measures the curvature of the reconstruction error in latent space:

$$G_{\text{eff}}(x) := J_g(z)^\top H_D(x) J_g(z). \tag{11}$$

The *Physics Sensitivity Metric* $W_{\text{eff}}$ measures the curva-

---

[1]For example, if $D_X(x, \hat{x}) = \|x - \hat{x}\|_2^2$, then $H_D(x) = 2I$.

ture of the physical observable loss in latent space:

$$W_{\text{eff}}(x) := J_g(z)^\top \underbrace{J_Q(x)^\top H_\ell(x) J_Q(x)}_{\text{Observable Stiffness}} J_g(z). \quad (12)$$

The metrics $G_{\text{eff}}$ and $W_{\text{eff}}$ determine how the covariance of latent perturbations contributes to expected distortion. Specifically, for a zero-mean perturbation satisfying $\mathbb{E}[\eta\eta^\top] = \Sigma$, the expected quadratic loss is given by a trace pairing between the relevant metric and $\Sigma$.

**Proposition 4.3** (Second-order distortion surrogates). *Under the linearisations* (9)–(10) *and the local quadratic approximations, the $\Sigma$-dependent parts of the expected distortions satisfy*

$$\mathbb{E}\Big[D_X\big(X, \hat{X}\big) \mid X = x\Big] \approx \text{Tr}(G_{\text{eff}}(x)\,\Sigma),$$

$$\mathbb{E}\Big[D_Q\big(Q(X), Q(\hat{X})\big) \mid X = x\Big] \approx \text{Tr}(W_{\text{eff}}(x)\,\Sigma).$$

Full details of the derivation are given in Appendix A.3.

---

**Example**

Power spectra and band energies are common quantities of interest in turbulence, seismology, and communications. Let $y_j$ be the frequency coefficients of an orthonormal transform (e.g, DFT or DCT). For frequency bands $B_k$, define the log-band energies

$$u_k(x) = \log e_k(x), \qquad e_k(x) = \sum_{j \in B_k} y_j^2.$$

Consider a quadratic observable loss $D_Q(u, \hat{u}) = \sum_{k=1}^K \sigma_k^{-2}(u_k - \hat{u}_k)^2$. Writing $J_{e_k}(x)$ for the Jacobian of the scalar band energy $e_k$, the induced physics sensitivity takes the schematic form

$$J_Q(x)^\top H_\ell J_Q(x) \sim \sum_{k=1}^K \frac{1}{\sigma_k^2 \, e_k(x)^2} J_{e_k}(x)^\top J_{e_k}(x).$$

The factor $1/e_k(x)^2$ acts as a gain control: preserving *log*-energies corresponds to controlling *relative* band-energy errors. As a result, the observable stiffness weights absolute energy perturbations more strongly in low-energy bands and more weakly in high-energy bands.

---

### 4.4. A local rate–distortion law

We now introduce our central characterisation of physics-aware compression, using the three quadratic objects in latent space derived previously.

Fix an operating point $x$ with $z = f_\phi(x)$. We interpret

$(\varepsilon_Q, \varepsilon_X)$ as local budgets for the second-order approximations of $D_Q$ and $D_X$, respectively.

---

**Theorem 4.4** (Local tangent-space rate–distortion law). *Under the perturbation model* (4) *and the local approximations of Propositions 4.1 and 4.3, the physics-aligned rate–distortion problem admits the local surrogate*

$$R_{\text{sur}}(\varepsilon_Q, \varepsilon_X; x) := \min_{\Sigma \succ 0} \quad \text{Tr}(H_R(z)\Sigma) - \log \det \Sigma \quad (13)$$

$$\text{s.t.} \quad \text{Tr}(W_{\text{eff}}(x)\Sigma) \leq \varepsilon_Q, \quad (14)$$

$$\text{Tr}(G_{\text{eff}}(x)\Sigma) \leq \varepsilon_X. \quad (15)$$

*If the budgets are feasible, the problem has a unique minimiser. Moreover, there exist Lagrange multipliers $\alpha \geq 0$ and $\gamma \geq 0$ such that*

$$(\Sigma^\star(x))^{-1} = H_R(z) + \alpha W_{\text{eff}}(x) + \gamma G_{\text{eff}}(x). \quad (16)$$

---

The surrogate problem follows by substituting the local rate expansion from Proposition 4.1 and the trace distortion surrogates from Proposition 4.3 into the constrained rate–distortion objective. The objective is strictly convex in $\Sigma$, since the trace term is linear and $-\log \det \Sigma$ is strictly convex on $\Sigma \succ 0$. The inverse-covariance expression follows from Karush–Kuhn–Tucker conditions, with full details in Appendix A.4.

### 4.5. Interpretation of physical alignment

Equation (16) shows that the optimal allocation is additive in precision:

$$(\Sigma^\star)^{-1} = \underbrace{H_R}_{\text{rate curvature}} + \alpha \underbrace{W_{\text{eff}}}_{\text{observable sensitivity}} + \gamma \underbrace{G_{\text{eff}}}_{\text{signal fidelity}}.$$

This gives a geometric inverse-stiffness interpretation of the covariance. Latent directions with large curvature under $H_R$ are costly to encode, directions with large curvature under $W_{\text{eff}}$ produce large observable-space distortion, and directions with large curvature under $G_{\text{eff}}$ produce large signal-space distortion. The optimal covariance therefore assigns lower variance to directions that are strongly penalised by these metrics.

A useful way to separate information cost from physical and signal sensitivity is to work in coordinates normalised by the rate metric (also called *rate whitening*). Assuming $H_R \succ 0$, define the rate-whitened metrics

$$\widetilde{W} := H_R^{-1/2} W_{\text{eff}} H_R^{-1/2}, \qquad \widetilde{G} := H_R^{-1/2} G_{\text{eff}} H_R^{-1/2}. \quad (17)$$

Then (16) can be written as

$$\Sigma^\star = H_R^{-1/2}\Big(I + \alpha \widetilde{W} + \gamma \widetilde{G}\Big)^{-1} H_R^{-1/2}. \quad (18)$$

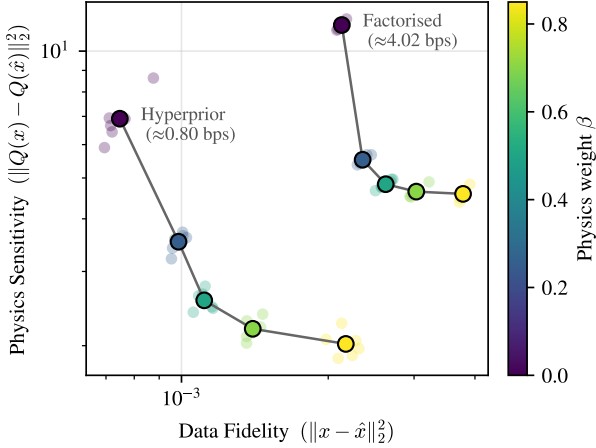

*Figure 4.* Data and observable space errors for different variational autoencoder models with hyperprior or factorised entropy model. Multiple repetitions are trained for varying physics weight $\beta$, tracing the Pareto frontier.

In these coordinates, $\widetilde{W}$ and $\widetilde{G}$ measure observable and signal sensitivity per unit rate cost. Physical alignment is therefore determined by the eigendirections of these rate-normalised metrics: if the dominant directions of $\widetilde{W}$ and $\widetilde{G}$ coincide, physical and signal fidelity suppress uncertainty along similar latent directions; if they differ, the two objectives compete for different directions of precision.

The same allocation rule also admits a linear-filtering interpretation. It is equivalent to the posterior covariance of a linear-Gaussian estimator with prior precision $H_R$ and observation precisions $\alpha W_{\text{eff}}$ and $\gamma G_{\text{eff}}$. This connects physics-aware compression to the denoising view of autoencoders, where compressed representations are encouraged to retain task-relevant signal structure (Vincent et al., 2008). In our setting, the denoising geometry is explicit: relative to the local rate metric, latent directions with larger observable or signal sensitivity receive smaller variance. Appendix A.6 gives the full derivation of the equivalence.

## 5. Consequences of Physical Alignment

### 5.1. A Physical No-Free-Lunch

Figure 4 illustrates the fixed-rate trade-off that motivates the alignment analysis: changing the physics weight $\beta$ reallocates error between the primary field and the observable. The local theory identifies a geometric condition under which this trade-off is unavoidable. The relevant comparison is between the rate-normalised physics and fidelity metrics introduced in (17). If these metrics favour different latent directions, then reducing observable-space distortion at fixed rate requires allocating precision away from directions preferred by the signal-fidelity objective.

**Definition 5.1** (Rate-whitened alignment). Assume $H_R \succ 0$ and define $\widetilde{W}, \widetilde{G}$ as in (17). We say that the task and fidelity are *rate-aligned* if $\widetilde{W}$ and $\widetilde{G}$ commute (admit a common eigenbasis). Otherwise they are *not rate-aligned*.

The following result formalises the fixed-rate consequence of this mismatch.

**Theorem 5.2** (No free lunch on physics fidelity at fixed rate). *Assume $H_R \succ 0$ and consider the rate-feasible set*

$$\mathcal{F}_\rho := \left\{ \Sigma \succ 0 : R_{\text{sur}}(\Sigma) \le \rho \right\}, \qquad (19)$$

*for some budget $\rho > 0$, where $R_{\text{sur}}$ is defined in (8). Define the optimal covariances at rate $\rho$ as*

$$\Sigma_{\text{Phys}} := \arg \min_{\Sigma \in \mathcal{F}_\rho} \text{Tr}(W_{\text{eff}}\Sigma),$$

$$\Sigma_{\text{MSE}} := \arg \min_{\Sigma \in \mathcal{F}_\rho} \text{Tr}(G_{\text{eff}}\Sigma),$$

*and assume these minimisers are unique.*
*If the task and fidelity are* not *rate-aligned (Definition 5.1), then improving physics at fixed rate strictly worsens fidelity, in the sense that*

$$\text{Tr}(W_{\text{eff}}\Sigma_{\text{Phys}}) < \text{Tr}(W_{\text{eff}}\Sigma_{\text{MSE}})$$
$$\implies \text{Tr}(G_{\text{eff}}\Sigma_{\text{Phys}}) > \text{Tr}(G_{\text{eff}}\Sigma_{\text{MSE}}). \tag{20}$$

Theorem 5.2 identifies a geometric source of Pareto tension: if the (rate-whitened) metrics are not rate-aligned, any move along the rate-feasible efficient frontier toward better physics fidelity inevitably degrades MSE. Full details of the proof are given in Appendix A.5.

The theorem should be read as a sufficient condition for fixed-rate Pareto tension. Non-commutativity of $\widetilde{W}$ and $\widetilde{G}$ gives a rotational mismatch: the observable and fidelity objectives require precision in different latent eigendirections. This is enough to force a strict trade-off under the theorem assumptions. The converse is not implied. If $\widetilde{W}$ and $\widetilde{G}$ commute, then the rotational mismatch is absent, but the two objectives may still prefer different allocations through their eigenvalue profiles. Thus, fixed-rate trade-offs can arise either from rotational mismatch, where the relevant eigenspaces differ, or from spectral mismatch, where the eigenspaces are shared but weighted differently.

### 5.2. Measuring alignment between physics and fidelity

The previous result is inspiration for a local alignment score on whether observable fidelity and signal fidelity require precision in the same latent directions.

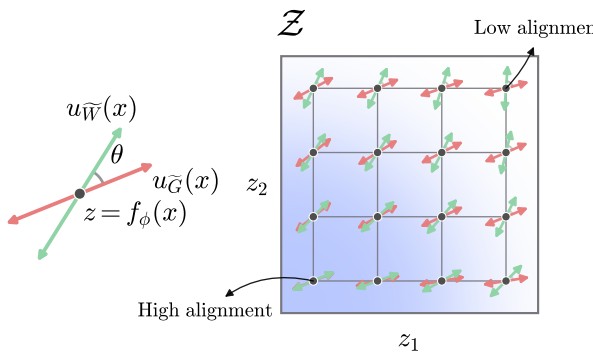

*Figure 5.* At each latent point $z = f_\phi(x)$, the red and green arrows denote the dominant fidelity-sensitive and physics-sensitive directions, respectively, in the rate-whitened geometry. Their acute angle $\theta(x)$ determines the local alignment score shown in the background.

---

**Definition 5.3** (Physical Alignment Score). Let $\widetilde{W}(x)$ and $\widetilde{G}(x)$ denote the rate-whitened physics and fidelity metrics at state $x$. Let $U_{W,k}(x) \in \mathbb{R}^{m \times k}$ (resp. $U_{G,k}(x)$) have orthonormal columns spanning the top-$k$ eigenspace of $\widetilde{W}(x)$ (resp. $\widetilde{G}(x)$). We define the *k-physical alignment* at $x$ as

$$\text{Align}_k(x) := \frac{1}{k} \left\| U_{W,k}(x)^\top U_{G,k}(x) \right\|_F^2 \in [0,1]. \tag{21}$$

Equivalently, if $\{\theta_i(x)\}_{i=1}^k$ are the principal angles between the two subspaces, then $\text{Align}_k(x) = \frac{1}{k} \sum_{i=1}^k \cos^2 \theta_i(x)$. A dataset-level score is obtained by averaging: $\text{Align}_k := \mathbb{E}_X[\text{Align}_k(X)]$.

$\text{Align}_k(x)$ measures whether the *physics-stiff* directions (top eigenvectors of $\widetilde{W}$) coincide with the *fidelity-stiff* directions (top eigenvectors of $\widetilde{G}$) in the rate-whitened geometry (see Fig. 5). Values $\text{Align}_k \approx 1$ indicate that the leading observable-sensitive and fidelity-sensitive subspaces coincide in the rate-whitened geometry, whereas low values $\text{Align}_k < 1$ indicate a *rotational* mismatch between the two objectives, consistent with stronger physics–MSE tension at fixed rate.

In practice we never form $\widetilde{W}$ or $\widetilde{G}$ explicitly, but access them only through matrix–vector products implemented via automatic differentiation. The dominant $k$-dimensional eigenspaces are then approximated using a standard randomised range-finding procedure: a Gaussian test matrix is propagated through the operator using these matvecs, followed by a small number of subspace-iteration steps to sharpen separation of the leading eigendirections before orthonormalisation. Full implementation details are given in Appendix B.

### 5.3. Spectral rate advantage from anisotropic physics

Let $\tilde{w}_1 \geq \tilde{w}_2 \geq \cdots \geq \tilde{w}_m \geq 0$ be the eigenvalues of the rate-whitened physics metric $\widetilde{W}$. The allocation rule (18) shows that these eigenvalues determine the rate-normalised cost of controlling the observable: directions with large $\tilde{w}_i$ require increased precision, while directions with small $\tilde{w}_i$ can tolerate larger variance. When the spectrum decays rapidly, observable sensitivity is concentrated in a low-dimensional subspace. Spectral concentration can therefore produce a rate advantage relative to an isotropic fidelity constraint: for the same observable distortion, fewer rate-normalised directions need to be controlled. We illustrate the modal allocation in the commuting, rate-aligned case in Fig. 6, and report the corresponding rate–distortion behaviour for the running CFD example in Fig. 7.

## 6. Experiments

We evaluate the proposed framework with two aims: first, to test whether the local tangent-space model accurately describes the behaviour of trained codecs; and second, to determine whether the proposed alignment diagnostic explains the trade-offs induced by physics-aware compression. The main-text experiments focus on turbulent fluid dynamics, where the geometric predictions can be probed most directly. Additional datasets, observables, and cross-domain analyses, which include cosmological simulation and electron microscopy, are reported in the Appendix.

**Data and physical observable.** Our primary testbed is the turbulent Navier–Stokes subset of PDEBench, where we compress 2D velocity fields $\mathbf{v} = (v_x, v_y)$. Following Fig. 1, the physical observable is vorticity, $Q(\mathbf{v}) = \partial_y v_x - \partial_x v_y$, and the physics loss is an $\ell_2$ discrepancy in observable space, $D_Q(Q(x), Q(\hat{x})) = \|Q(x) - Q(\hat{x})\|_2^2$. Signal-space fidelity is measured by MSE (and reported as PSNR when plotting rate–distortion curves). Full dataset descriptions, additional observables, preprocessing, and implementation details are deferred to Appendix C.

Models are trained by minimising the rate-distortion Lagrangian with a *physics weight* $\beta \in [0, 1]$:

$$\min_{\phi, \theta, \psi} \mathcal{L}_{\text{train}} := \mathbb{E}_{X \sim p(x)} \begin{bmatrix} R_\psi(\hat{Z}) + \lambda(1 - \beta) D_X(X, \hat{X}) \\ + \lambda \beta D_Q(Q(X), Q(\hat{X})) \end{bmatrix},$$
$$Z = f_\phi(X), \qquad \hat{Z} = \mathcal{Q}(Z), \qquad \hat{X} = g_\theta(\hat{Z}). \tag{22}$$

where $R_\psi(\hat{Z})$ is the variational rate proxy. By construction, $\beta = 0$ corresponds to MSE-only training and $\beta = 1$ to physics-only training.

**Validating the sample-local geometric model.** To test the core approximation behind our theory, we probe trained

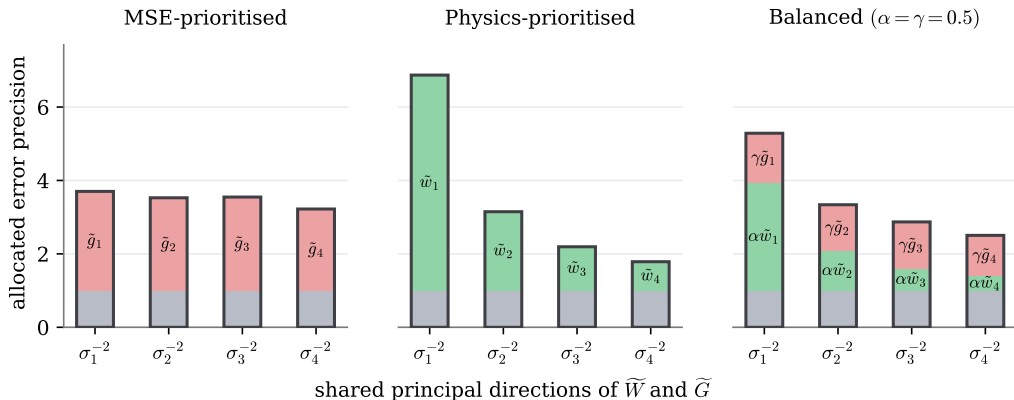

*Figure 6.* Assuming $\widetilde{W}$ and $\widetilde{G}$ share a common eigenbasis, the local allocation rule decomposes by mode, with $(\tilde{\sigma}_i^\star)^{-2} = 1 + \alpha\widetilde{w}_i + \gamma\widetilde{g}_i$. Each stacked bar shows the corresponding combined precision: gray is the rate baseline (the 1 constant), green the physics contribution $(\alpha\widetilde{w}_i)$, and red the fidelity contribution $(\gamma\widetilde{g}_i)$. The three panels compare MSE-prioritised, physics-prioritised, and balanced allocations.

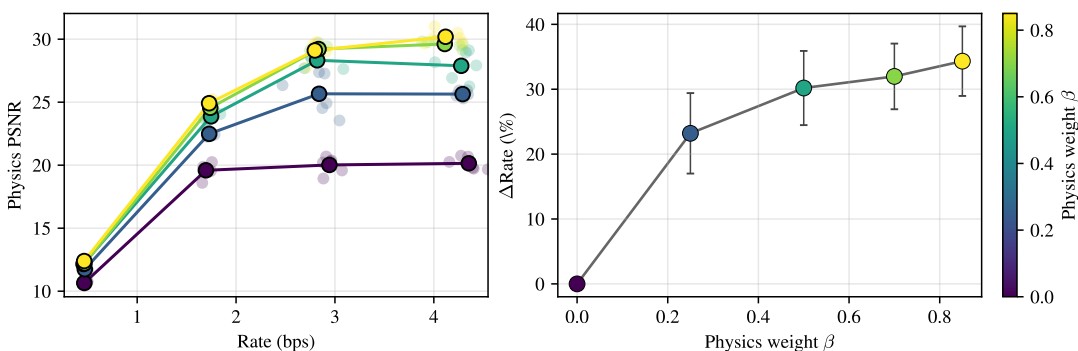

*Figure 7.* Rate–distortion curves for a hyperprior model for the physics loss (left) and rate gain (right), computed as the average rate difference over a common PSNR range (Bjontegaard, 2001).

models locally in latent space. For random validation samples $x$, we encode $z = f_\phi(x)$ and inject controlled perturbations $\eta$ with prescribed covariance (diagonal and full-rank variants), forming $\hat{z} = z + \eta$ and decoding $\hat{x} = g_\theta(\hat{z})$. We then compare (i) the change in variational rate against the quadratic surrogate in Lemma 4.1, and (ii) the change in signal/physics losses against the trace surrogates in Proposition 4.3. We quantify agreement via the coefficient of determination ($R^2$) over a sweep of perturbation magnitudes. Fig. 8 shows that the second-order models remain predictive for moderate noise levels, supporting the use of $H_R$, $W_{\text{eff}}$, and $G_{\text{eff}}$ as local geometry descriptors. Appendix D reports additional low-bitrate experiments.

**Rate–distortion–physics trade-offs at fixed rate.** Next we examine the empirical Pareto frontier induced by $\beta$. For each target bitrate, we train multiple models with different $\beta$ and evaluate both signal distortion $D_X$ and observable distortion $D_Q$. Fig. 4 visualises this frontier: increasing $\beta$ improves vorticity fidelity while typically degrading MSE, consistent with Theorem 5.2 when the (rate-whitened) physics

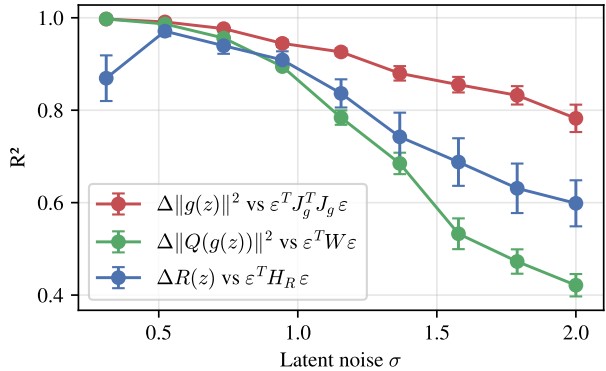

*Figure 8.* Averaged $R^2$ statistic between the quadratic approximations and their counterparts. Values are averaged over various data samples, models, and hyperparameter configurations.

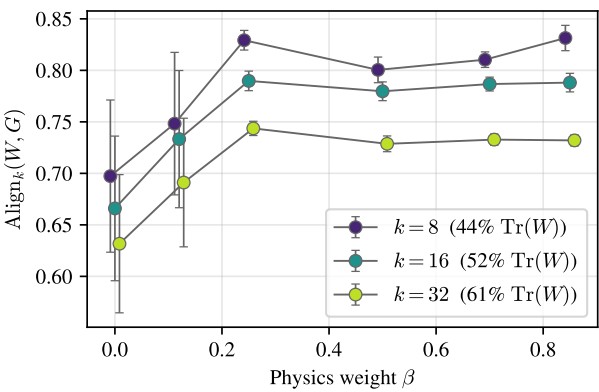

*Figure 9.* $\mathrm{Align}_k$ metric for $k = 8, 16, 32$, averaged over data samples, on a hyperprior model. For every $k$, the percentage of trace coverage is indicated.

and fidelity metrics are not aligned. Importantly, this behavior reflects *reallocation* of error into directions that are weakly sensed by $Q$ rather than a uniform improvement. This is also observed in the additional physical observables included in Appendix D.

**Spectral alignment as an explanatory diagnostic.** Finally, we connect the observed trade-offs and rate gains to the rate-whitened geometry (18). We compute the physical alignment score $\mathrm{Align}_k$ (Definition 5.3) using only matrix–vector products with the implicit operators $\widetilde{W}$ and $\widetilde{G}$, avoiding explicit formation of dense matrices. Fig. 9 shows that $\mathrm{Align}_k$ increases systematically with $\beta$, indicating that physics-aware training reshapes the representation so that physics-stiff directions increasingly overlap with fidelity-stiff directions in the rate-whitened geometry. This trend helps explain when improved physics fidelity is achieved with mild MSE degradation, and it is consistent with the rate advantages observed in Fig. 7 when sensitivity is spectrally concentrated (Section 5.3). Appendix D shows that the diagnostic remains informative across additional domains and observables.

## 7. Limitations

- **Alignment is described, not enforced.** While the theory explains how physical alignment relates to trade-offs and rate efficiency, a full analysis of how training objectives and regularisers help improve the alignment is left for future work.

- **Reconstruction bias is not modelled explicitly.** The analysis treats compression through local latent perturbations around an operating point, so it captures uncertainty-induced error rather than systematic encoder–decoder bias from finite model capacity or imperfect optimisation.

## 8. Conclusions

This paper gives a local geometric explanation of physics-aware learned compression. The central insight is that preserving a physical observable and preserving standard fidelity each privilege certain latent directions, and that the alignment of these directions determines whether physics-aware training yields a tradeoff. Misalignment explains the fixed-rate physics–MSE tension often seen in practice, while spectral concentration explains when preserving physics can remain efficient.

We made this precise through a local tangent-space rate–distortion law, an explicit inverse-stiffness allocation rule for compression noise, and a practical diagnostic based on dominant eigenspace overlap. Experiments across several scientific domains support the theory and show that the diagnostic is predictive in practice.

**Take-aways**

- **Physics-aware compression has local geometry.** At each latent operating point, rate, observable sensitivity, and signal fidelity induce a local allocation law for compression noise.

- **Misalignment implies a fixed-rate trade-off.** Under the local model, when physical observables and fidelity metrics are not aligned, improving observable fidelity at fixed rate worsens standard distortion.

- **Alignment can be estimated in practice.** The physical alignment score provides a computable diagnostic for assessing physics–fidelity trade-offs.

- **Anisotropy can yield rate advantages.** Physics-aware compression can be rate-efficient when observable sensitivity concentrates in a low-dimensional subspace.

## Acknowledgements

Aleix received the support of a fellowship from the "la Caixa" Foundation (ID 100010434), under fellowship code LCF/BQ/PFA25/110000. We thank Ryan Cherian for the relevant discussion during the development of the work.

## Impact Statement

This paper presents work whose goal is to advance the field of Machine Learning. There are many potential societal consequences of our work, none which we feel must be specifically highlighted here.

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

# A. Additional details for proofs

## A.1. Proof of variational rate bounds

We prove the chain of inequalities $I(X; \hat{X}) \leq I(X; \hat{Z}) \leq R_{\text{var}}$.

**Step 1: Data processing $I(X; \hat{X}) \leq I(X; \hat{Z})$.** By assumption $\hat{X} = g_\theta(\hat{Z})$ for a measurable decoder $g_\theta$, hence $X \to \hat{Z} \to \hat{X}$ is a Markov chain. By the data processing inequality,

$$I(X; \hat{X}) \leq I(X; \hat{Z}).$$

**Step 2: Variational upper bound $I(X; \hat{Z}) \leq R_{\text{var}}$.** Let $q(\hat{z}) = \int q(\hat{z} \mid x) p(x)\, dx$ be the induced marginal under $p(x)q(\hat{z} \mid x)$. We expand the variational rate:

$$R_{\text{var}} = \mathbb{E}_{p(x)}\left[\text{KL}\Big(q(\hat{Z} \mid X) \,\|\, p_\psi(\hat{Z})\Big)\right]$$

$$= \mathbb{E}_{p(x)}\mathbb{E}_{q(\hat{z}|x)}\big[\log q(\hat{z} \mid x) - \log p_\psi(\hat{z})\big].$$

Add and subtract $\log q(\hat{z})$ inside the expectation:

$$R_{\text{var}} = \mathbb{E}_{p(x)}\mathbb{E}_{q(\hat{z}|x)}\big[\log q(\hat{z} \mid x) - \log q(\hat{z})\big] + \mathbb{E}_{p(x)}\mathbb{E}_{q(\hat{z}|x)}\big[\log q(\hat{z}) - \log p_\psi(\hat{z})\big]$$

$$= I_q(X; \hat{Z}) + \text{KL}\Big(q(\hat{Z}) \,\|\, p_\psi(\hat{Z})\Big)$$

$$\geq I_q(X; \hat{Z}),$$

since KL divergence is nonnegative. Identifying $I_q(X; \hat{Z})$ with $I(X; \hat{Z})$ under the joint $p(x)q(\hat{z} \mid x)$ yields $I(X; \hat{Z}) \leq R_{\text{var}}$, completing the proof.

## A.2. Proof of Proposition 4.1

We evaluate the variational rate at a fixed input $x$ with latent representation $z = z(x)$. The variational posterior is given by the local Gaussian channel $q_\Sigma(\hat{z} \mid x) = \mathcal{N}(\hat{z}; z, \Sigma)$. By definition:

$$R_{\text{var}}(x) = \underbrace{\mathbb{E}_{q_\Sigma}[\log q_\Sigma(\hat{z} \mid x)]}_{\text{(i)}} - \underbrace{\mathbb{E}_{q_\Sigma}[\log p_\psi(\hat{z})]}_{\text{(ii)}}. \tag{23}$$

Let $\hat{z} = z + \eta$, where $\eta \sim \mathcal{N}(0, \Sigma)$. We analyze the two terms separately.

**(i) Negative Entropy:** The first term is the negative differential entropy of a multivariate Gaussian $\mathcal{N}(z, \Sigma)$ in $m$ dimensions. Using the standard entropy formula:

$$\mathbb{E}_{q_\Sigma}[\log q_\Sigma(\hat{z} \mid x)] = -H(q_\Sigma) = -\frac{1}{2}\log \det(2\pi e \Sigma) = -\frac{1}{2}\log \det \Sigma + C_1, \tag{24}$$

where $C_1$ collects terms independent of $\Sigma$.

**(ii) Expected Prior Log-Likelihood:** For the second term, we apply the local expansion of the prior log-likelihood assumed in (6) around the center $z$:

$$-\log p_\psi(z + \eta) = -\log p_\psi(z) + \nabla[-\log p_\psi(z)]^\top \eta + \frac{1}{2}\eta^\top H_R(z)\eta + o(\|\eta\|^2). \tag{25}$$

Taking the expectation with respect to $\eta \sim \mathcal{N}(0, \Sigma)$: The constant term $-\log p_\psi(z)$ remains unchanged; the linear term vanishes because $\mathbb{E}[\eta] = 0$; the quadratic term becomes $\frac{1}{2}\mathbb{E}[\eta^\top H_R(z)\eta] = \frac{1}{2}\text{Tr}(H_R(z)\Sigma)$ using the cyclic property of the trace.

Thus,

$$-\mathbb{E}_{q_\Sigma}[\log p_\psi(\hat{z})] = \text{const.} + \frac{1}{2}\text{Tr}(H_R(z)\Sigma) + o(\|\Sigma\|). \tag{26}$$

Combining (i) and (ii) yields the local expansion:

$$R_{\text{var}}(x) = \text{const.} + \frac{1}{2}\text{Tr}(H_R(z)\Sigma) - \frac{1}{2}\log \det \Sigma + o(\|\Sigma\|). \tag{27}$$

Averaging over $p(x)$ preserves this form up to an additive constant, proving the lemma.

### A.3. Proof of Proposition 4.3

We analyze the expected distortion induced by the local perturbation $\hat{z} = z + \eta$, where $\eta$ has zero mean and covariance $\Sigma$.

**Linearisation.** Let $J_g(z)$ be the Jacobian of the decoder $g_\theta$ at $z$. Linearising the decoder around $z$, the deviation in the data space is:

$$\delta x := \hat{x} - x \approx g_\theta(z) + J_g(z)\eta - x \approx J_g(z)\eta,$$

where we assume the deterministic reconstruction error is negligible or absorbed into higher-order terms.

**Quadratic Expectation.** We rely on a standard result for quadratic expansions under zero-mean noise. For any smooth loss $\mathcal{L}(u, \hat{u})$ minimised at $\hat{u} = u$, the second-order expansion with deviation $\delta = A\eta$ satisfies:

$$\mathbb{E}[\mathcal{L}(u, u + \delta)] = \text{const.} + \frac{1}{2}\mathbb{E}\big[\eta^\top A^\top H_\mathcal{L}(u)A\eta\big] + o(\|\eta\|^2) \approx \text{const.} + \frac{1}{2}\text{Tr}\big(A^\top H_\mathcal{L}(u)A\Sigma\big),$$

where $H_\mathcal{L}$ is the Hessian with respect to the second argument. Throughout the main text we absorb the universal factor $1/2$ into the normalisation of the local distortion budgets.

**Signal-Space Distortion ($D_X$).** Using the general result with the mapping $A = J_g(z)$ and Hessian $H_D(x)$, we immediately obtain:

$$\mathbb{E}[D_X(x, \hat{x})] \approx \text{const.} + \frac{1}{2}\text{Tr}\big(J_g(z)^\top H_D(x)J_g(z)\Sigma\big). \tag{28}$$

Defining $G_{\text{eff}}(x) = J_g(z)^\top H_D(x)J_g(z)$ recovers the result.

**Physics Loss ($D_Q$).** The observable map linearises as $Q(\hat{x}) \approx Q(x) + J_Q(x)\delta x$. The effective linear mapping from noise $\eta$ to observable deviation is therefore $A = J_Q(x)J_g(z)$. Applying the general result with Hessian $H_\ell(x)$:

$$\mathbb{E}[D_Q] \approx \text{const.} + \frac{1}{2}\text{Tr}\big((J_Q J_g)^\top H_\ell(J_Q J_g)\Sigma\big). \tag{29}$$

Defining $W_{\text{eff}}(x) = J_g(z)^\top J_Q(x)^\top H_\ell(x)J_Q(x)J_g(z)$ recovers the result.

### A.4. Proof of Theorem 4.4

We prove that the solution to (13)–(15) is $\Sigma^\star = (H_R + \alpha W_{\text{eff}} + \gamma G_{\text{eff}})^{-1}$ for suitable multipliers $\alpha, \gamma \geq 0$.

**Step 1: Convexity and existence/uniqueness.** The feasible set $\{\Sigma \succ 0 : \text{Tr}(W_{\text{eff}}\Sigma) \leq \varepsilon_Q, \ \text{Tr}(G_{\text{eff}}\Sigma) \leq \varepsilon_X\}$ is convex because the constraints are linear in $\Sigma$. The objective $f(\Sigma) = \text{Tr}(H_R\Sigma) - \log \det \Sigma$ is strictly convex over $\Sigma \succ 0$ because $\text{Tr}(H_R\Sigma)$ is linear and $-\log \det \Sigma$ is strictly convex. Hence the problem admits at most one minimiser; under Slater's condition (feasible interior), a minimiser exists and KKT conditions are necessary and sufficient.

**Step 2: Lagrangian and stationarity.** Form the Lagrangian

$$\mathcal{L}(\Sigma, \alpha, \gamma) = \text{Tr}(H_R\Sigma) - \log \det \Sigma + \alpha\big(\text{Tr}(W_{\text{eff}}\Sigma) - \varepsilon_Q\big) + \gamma\big(\text{Tr}(G_{\text{eff}}\Sigma) - \varepsilon_X\big),$$

with $\alpha, \gamma \geq 0$. The derivative identities

$$\nabla_\Sigma \text{Tr}(A\Sigma) = A, \qquad \nabla_\Sigma(-\log \det \Sigma) = -\Sigma^{-1}$$

yield the stationarity condition

$$0 = \nabla_\Sigma \mathcal{L} = H_R - \Sigma^{-1} + \alpha W_{\text{eff}} + \gamma G_{\text{eff}}.$$

Rearranging gives

$$\Sigma^{-1} = H_R + \alpha W_{\text{eff}} + \gamma G_{\text{eff}},$$

and therefore

$$\Sigma^\star = \big(H_R + \alpha W_{\text{eff}} + \gamma G_{\text{eff}}\big)^{-1}.$$

**Step 3: Complementary slackness and feasibility.** Together with primal feasibility and dual feasibility, the remaining KKT conditions are

$$\alpha \geq 0, \ \gamma \geq 0, \qquad \mathrm{Tr}(W_{\mathrm{eff}}\Sigma^{\star}) \leq \varepsilon_Q, \ \mathrm{Tr}(G_{\mathrm{eff}}\Sigma^{\star}) \leq \varepsilon_X,$$

and complementary slackness:

$$\alpha\big(\mathrm{Tr}(W_{\mathrm{eff}}\Sigma^{\star}) - \varepsilon_Q\big) = 0, \qquad \gamma\big(\mathrm{Tr}(G_{\mathrm{eff}}\Sigma^{\star}) - \varepsilon_X\big) = 0.$$

Since the problem is strictly convex, any $(\Sigma^{\star}, \alpha, \gamma)$ satisfying KKT is the unique global optimum. This completes the proof.

### A.5. Proof of Theorem 5.2

Fix a surrogate rate budget $\rho$. The proof proceeds in three steps: (1) simplified coordinates (whitening), (2) a general optimisation lemma, and (3) establishing the trade-off via contradiction.

**Whitening the Prior.** To simplify the rate constraint, we work in a coordinate system whitened by the prior curvature $H_R$. Assume $H_R \succ 0$ and define the transformed variables:

$$\widetilde{\Sigma} := H_R^{1/2}\Sigma H_R^{1/2}, \quad \widetilde{W} := H_R^{-1/2}W_{\mathrm{eff}}H_R^{-1/2}, \quad \widetilde{G} := H_R^{-1/2}G_{\mathrm{eff}}H_R^{-1/2}. \tag{30}$$

In these coordinates, the variational rate $R_{\mathrm{sur}}$ simplifies to an isotropic form (up to a constant independent of $\Sigma$):

$$\widetilde{R}(\widetilde{\Sigma}) = \frac{1}{2}\mathrm{Tr}(\widetilde{\Sigma}) - \frac{1}{2}\log\det\widetilde{\Sigma}.$$

Consequently, the feasible set becomes $\widetilde{\mathcal{F}}_{\tilde{\rho}} = \{\widetilde{\Sigma} \succ 0 : \widetilde{R}(\widetilde{\Sigma}) \leq \tilde{\rho}\}$. Furthermore, the trace objectives are invariant under whitening: $\mathrm{Tr}(W_{\mathrm{eff}}\Sigma) = \mathrm{Tr}(\widetilde{W}\widetilde{\Sigma})$. Thus, it suffices to prove the theorem in the whitened domain.

**General Optimisation Result.** Consider minimising a linear objective $\mathrm{Tr}(A\widetilde{\Sigma})$ with $A \succeq 0$ over the feasible set $\widetilde{\mathcal{F}}_{\tilde{\rho}}$.

**Lemma A.1.** *The unique minimiser $\widetilde{\Sigma}^*$ of $\min_{\widetilde{\Sigma} \in \widetilde{\mathcal{F}}_{\tilde{\rho}}} \mathrm{Tr}(A\widetilde{\Sigma})$ satisfies the form:*

$$(\widetilde{\Sigma}^*)^{-1} = I + \lambda A, \quad \text{for some scalar } \lambda \geq 0. \tag{31}$$

*Proof.* The problem is strictly convex (as $-\log\det$ is strictly convex), ensuring a unique solution. The Lagrangian is $\mathcal{L}(\widetilde{\Sigma}, \mu) = \mathrm{Tr}(A\widetilde{\Sigma}) + \mu(\widetilde{R}(\widetilde{\Sigma}) - \tilde{\rho})$. The stationarity condition $\nabla_{\widetilde{\Sigma}}\mathcal{L} = 0$ yields:

$$A + \frac{\mu}{2}(I - \widetilde{\Sigma}^{-1}) = 0 \quad \implies \quad \widetilde{\Sigma}^{-1} = I + \frac{2}{\mu}A.$$

Setting $\lambda = 2/\mu \geq 0$ completes the lemma. $\qquad\square$

**Distinctness and Strict Inequalities.** Let $\widetilde{\Sigma}_{\mathrm{Phys}}$ and $\widetilde{\Sigma}_{\mathrm{MSE}}$ be the unique minimisers for the physics loss ($\widetilde{W}$) and MSE ($\widetilde{G}$) respectively. By the lemma, their inverses are:

$$\widetilde{\Sigma}_{\mathrm{Phys}}^{-1} = I + \lambda_W\widetilde{W}, \qquad \widetilde{\Sigma}_{\mathrm{MSE}}^{-1} = I + \lambda_G\widetilde{G}. \tag{32}$$

We argue by contradiction. Suppose there is no trade-off, i.e., the optimal covariances are identical ($\widetilde{\Sigma}_{\mathrm{Phys}} = \widetilde{\Sigma}_{\mathrm{MSE}}$). Then:

$$I + \lambda_W\widetilde{W} = I + \lambda_G\widetilde{G} \quad \implies \quad \widetilde{W} \propto \widetilde{G}.$$

This implies $\widetilde{W}$ and $\widetilde{G}$ are proportional (and thus commute), which violates the assumption that the task and fidelity are *not rate-aligned* (Definition 5.1). Therefore, $\widetilde{\Sigma}_{\mathrm{Phys}} \neq \widetilde{\Sigma}_{\mathrm{MSE}}$.

**Conclusion.** Since $\widetilde{\Sigma}_{\mathrm{Phys}}$ is the *unique* minimiser of the physics loss:

$$\mathrm{Tr}(\widetilde{W}\widetilde{\Sigma}_{\mathrm{Phys}}) < \mathrm{Tr}(\widetilde{W}\widetilde{\Sigma}_{\mathrm{MSE}}).$$

Mapping back to the original coordinates via the trace invariance established in Step 1 yields:

$$\mathrm{Tr}(W_{\mathrm{eff}}\Sigma_{\mathrm{Phys}}) < \mathrm{Tr}(W_{\mathrm{eff}}\Sigma_{\mathrm{MSE}}).$$

The reverse inequality for the MSE loss follows symmetrically. $\qquad\square$

### A.6. Physics Alignment as Linear-Gaussian Filtering

We can reinterpret the optimal error allocation not as a compression problem, but as a classical linear-Gaussian estimation problem. Suppose we wish to estimate the latent perturbation $\eta \in \mathbb{R}^m$ based on three independent sources of information: a prior belief and two noisy "measurements" corresponding to the physics and fidelity constraints.

**The Estimation Setup.** Let the "true" perturbation be $\eta$. We treat the rate constraint as a Gaussian prior centered at zero (reflecting the bit-cost of deviations):

$$\eta \sim \mathcal{N}(0, H_R^{-1}). \tag{33}$$

Next, we view the penalties on physics and signal fidelity as noisy linear observations. We define linearised observation operators $A$ and $B$ via the matrix square roots of the loss Hessians:

$$A := H_\ell^{1/2} J_Q(x) J_g(z), \tag{34}$$

$$B := H_D^{1/2} J_g(z). \tag{35}$$

We model the constraints as observing "zero error" targets with additive Gaussian white noise:

$$r_Q = A\eta + \xi_Q, \qquad \xi_Q \sim \mathcal{N}(0, \alpha^{-1}I), \tag{36}$$

$$r_X = B\eta + \xi_X, \qquad \xi_X \sim \mathcal{N}(0, \gamma^{-1}I). \tag{37}$$

Here, the noise variances $\alpha^{-1}$ and $\gamma^{-1}$ represent our tolerance for error in the physics and signal domains, respectively.

**The Posterior Precision.** We seek the posterior distribution $p(\eta \mid r_Q = 0, r_X = 0)$. By Bayes' rule, the log-posterior is the sum of the log-prior and log-likelihoods. Focusing on the quadratic terms (which determine the covariance):

$$-\log p(\eta \mid \cdot) = \underbrace{\frac{1}{2}\eta^\top H_R \eta}_{\text{Prior}} + \underbrace{\frac{\alpha}{2}\|A\eta\|_2^2}_{\text{Physics Likelihood}} + \underbrace{\frac{\gamma}{2}\|B\eta\|_2^2}_{\text{Fidelity Likelihood}} + (\text{linear in } \eta) + \text{const.} \tag{38}$$

$$= \frac{1}{2}\eta^\top \left( H_R + \alpha A^\top A + \gamma B^\top B \right) \eta + \ldots \tag{39}$$

Recognising that $A^\top A = W_{\text{eff}}$ and $B^\top B = G_{\text{eff}}$, we see that the posterior precision matrix is exactly the inverted covariance matrix from Theorem 4.4:

$$\Lambda_{\text{post}} = H_R + \alpha W_{\text{eff}} + \gamma G_{\text{eff}} = (\Sigma^\star)^{-1}. \tag{40}$$

**The Wiener Filter Interpretation.** This equivalence implies that the optimal physics-aligned codec shapes its quantisation noise $\Sigma^\star$ to match the *posterior uncertainty* of an ideal Bayesian estimator that has observed the physics and fidelity constraints. In the eigenbasis of the rate metric (where $H_R = I$), the variance along each mode $i$ shrinks according to the canonical Wiener/Ridge factor:

$$(\tilde{\sigma}_i^\star)^2 = \frac{1}{1 + \underbrace{\alpha \tilde{w}_i}_{\text{Physics SNR}} + \underbrace{\gamma \tilde{g}_i}_{\text{Fidelity SNR}}}. \tag{41}$$

This confirms that the method allocates bits (reduces variance) strictly in proportion to the "Signal-to-Noise Ratio" of the physical and signal constraints relative to the rate cost.

## B. Computing the physical alignment score

We estimate the alignment score from Section 5.2 without explicitly forming the dense rate-whitened operators $\widetilde{G}(x)$ and $\widetilde{W}(x)$. Algorithm 1 summarises the resulting procedure. Throughout this appendix we fix a latent point $z = f_\phi(x)$ and suppress the dependence on $x$.

The operators are accessed only through matrix–vector products. Starting from the pullback metrics of Definition 4.2, a whitened matvec has the form

$$v \mapsto \widetilde{G}v = H_R^{-1/2} G_{\text{eff}} \big( H_R^{-1/2} v \big), \qquad v \mapsto \widetilde{W}v = H_R^{-1/2} W_{\text{eff}} \big( H_R^{-1/2} v \big).$$

---

**Algorithm 1** Estimating $\text{Align}_k(x)$ at $z = f_\phi(x)$

---

**Require:** Matvecs for $G_{\text{eff}}$ and $W_{\text{eff}}$; rate metric $H_R$; rank $k$; oversampling $p$; subspace iterations $q$

    Set $\ell \leftarrow k + p$

    Build matvecs for $\widetilde{G}$ and $\widetilde{W}$ using $H_R^{-1/2}$

    **for** $A \in \{\widetilde{G}, \widetilde{W}\}$ **do**

        Draw $\Omega \in \mathbb{R}^{n \times \ell}$ with i.i.d. Gaussian entries

        $Y \leftarrow A\Omega$

        **for** $t = 1, \ldots, q$ **do**

            $Q \leftarrow \text{orth}(Y)$

            $Y \leftarrow AQ$

        **end for**

        $Q \leftarrow \text{orth}(Y)$

        $B \leftarrow \frac{1}{2}(Q^\top A Q + Q^\top A^\top Q)$

        Compute $B = V_A \Lambda_A V_A^\top$

        $U_{A,k} \leftarrow Q V_A(:, 1{:}k)$

    **end for**

    **return** $\text{Align}_k(x) = k^{-1} \|U_{W,k}^\top U_{G,k}\|_F^2$

---

The products with $G_{\text{eff}}$ and $W_{\text{eff}}$ are computed using Jacobian–vector and vector–Jacobian products, following standard directional second-order automatic differentiation (Pearlmutter, 1994). For high-dimensional latent spaces, $H_R$ may be replaced by a diagonal approximation in the whitening factors.

For each operator $A \in \{\widetilde{G}, \widetilde{W}\}$, we estimate its dominant $k$-dimensional eigenspace with randomised range finding (Halko et al., 2011). We draw a Gaussian test matrix $\Omega \in \mathbb{R}^{n \times \ell}$, with $\ell = k + p$, form $Y = A\Omega$, and apply $q$ subspace-iteration steps. After orthonormalising the resulting basis $Q$, we solve the small eigenproblem $B = Q^\top A Q$ and lift the leading eigenvectors back to the latent space. This gives bases $U_{G,k}$ and $U_{W,k}$, from which

$$\text{Align}_k(x) = \frac{1}{k} \left\| U_{W,k}^\top U_{G,k} \right\|_F^2$$

is the average squared cosine of the principal angles between the two subspaces (Björck & Golub, 1973). For computing diagnostics (i.e., trace coverage of the top-$k$ eigenspace) we use Hutchinson trace estimates (Hutchinson, 1989).

## C. Experimental Setup and Additional Details

This section summarises the datasets, observables, preprocessing steps, and training details used in our experiments.

### C.1. Domains, compressed fields, and observables

We evaluate the proposed framework across three scientific domains chosen to span different spatial structure, observables, and downstream priorities: turbulent fluid dynamics, cosmological simulations, and electron microscopy. For each domain, we specify both the compressed field and the observable used to define the physics-aware objective. Data samples are shown in Fig. 10.

**Turbulent Fluid Dynamics (PDEBench).** We use data from **PDEBench** (Takamoto et al., 2022), focusing on the 2D compressible Navier–Stokes equations in a turbulent regime. The flow is nearly incompressible ($M = 0.1$), while the very small shear and bulk viscosities ($\eta, \zeta = 10^{-8}$) induce a high-Reynolds-number regime with fine-scale vortical structure. We compress the velocity field $\mathbf{v} = (v_x, v_y)$.

The main observable used in the experiments of Section 6 is the **vorticity**,

$$\omega(\mathbf{v}) = \partial_y v_x - \partial_x v_y,$$

which is highly sensitive to local compression artefacts because it depends on spatial derivatives. In the additional appendix

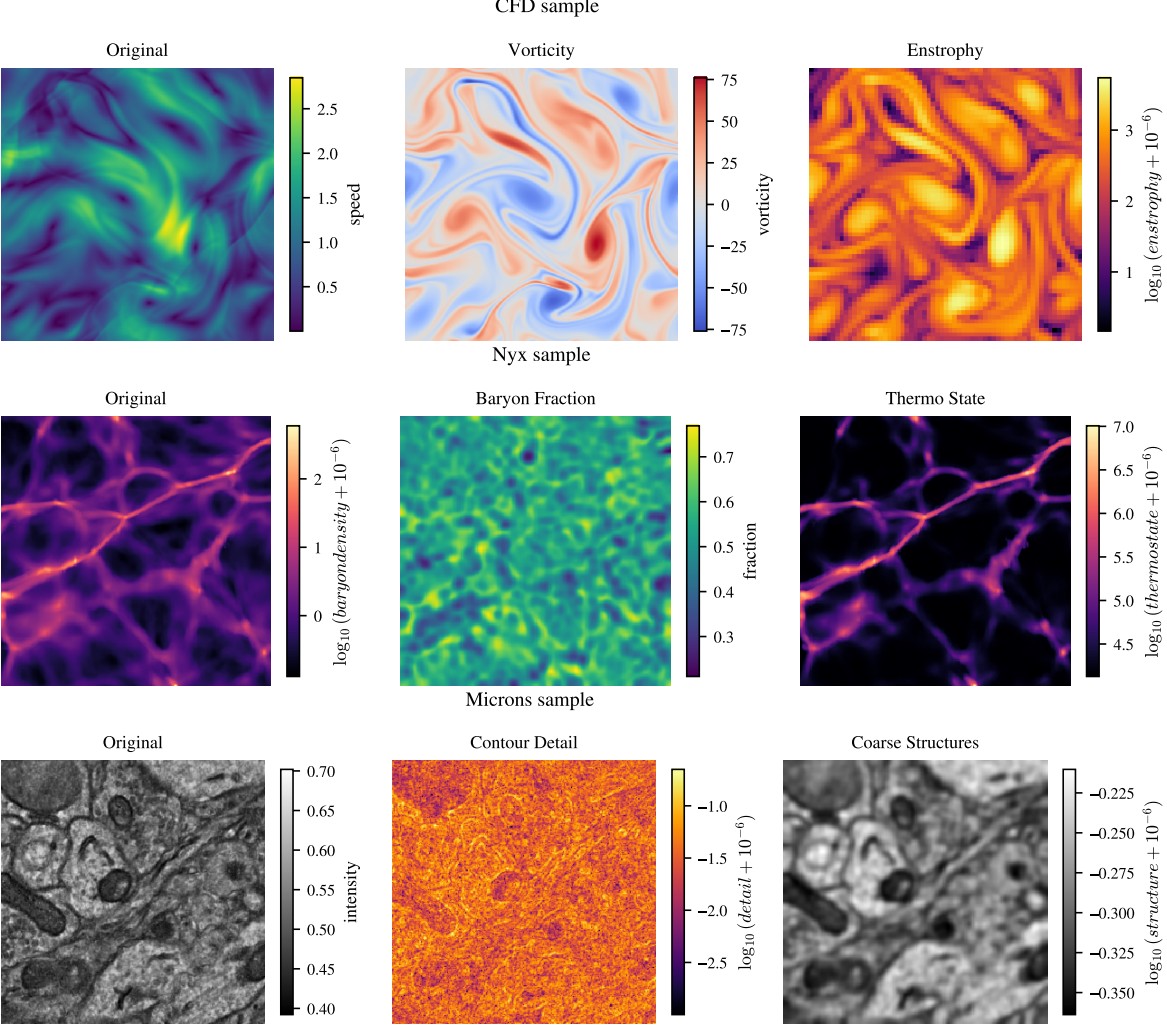

*Figure 10.* A sample for every data source: original (left) and two observables (middle and right).

experiments, we also consider a **windowed enstrophy** observable,

$$q_{\text{ens}}(x) = \frac{1}{|W|} \sum_{\mathbf{v} \in W(x)} \omega(\mathbf{v})^2,$$

which emphasises localised rotational activity while smoothing pixel-scale fluctuations, hence carrying meaning about larger scale energy flows.

**Nyx cosmological simulations.** We use slices from **Nyx**, a massively parallel cosmological hydrodynamics code designed to simulate baryonic gas and dark matter on large scales (Almgren et al., 2013). In this domain, we consider observables derived from baryon density, dark-matter density, and temperature.

Our first observable is the **local baryon fraction**,

$$f_b = \frac{\rho_b}{\rho_b + \rho_{dm} + \varepsilon},$$

which emphasises local compositional structure. Our second observable is a **thermodynamic-state proxy**, defined from

$$p = \rho_b T, \qquad s = T \rho_b^{-(\gamma-1)}, \qquad q_{\text{thermo}} = p + s,$$

which highlights thermodynamic regimes, especially hot and dense regions.

**Electron microscopy of mouse visual cortex.** We also evaluate on grayscale electron microscopy (EM) imagery from the **MICrONS / minnie65** cerebral cortex volume. This domain provides a qualitatively different setting in which the relevant structure is morphological rather than fluid or thermodynamic.

We consider two observables. The first is a high-frequency **contour-feature map**,

$$q_{\text{contour}}(I) = \sqrt{(w_x \partial_x I)^2 + (w_y \partial_y I)^2 + \varepsilon} + \lambda |\Delta I|,$$

which emphasises membranes, thin boundaries, and fine structural edges. The second is a low-frequency **coarse-structure field**,

$$q_{\text{coarse}}(I) = G_\sigma * I,$$

which suppresses edge-level detail and retains larger-scale morphology and contrast structure.

### C.2. Implementation and Training Details

To ensure rigorous comparison, we conduct extensive hyperparameter sweeps across architectures, rate constraints, and physics-aware objectives.

**Architectures.** We evaluate two distinct classes of learned image compression architectures:

1. **Factorised Prior:** A standard convolutional autoencoder where the latent distribution is modeled as fully factorised (independent) non-parametric densities.

2. **Scale Hyperprior:** A hierarchical model that utilises side information (a hyper-latent) to predict the parameters of the Gaussian distribution of the main latent code, capturing spatial dependencies more effectively (Ballé et al., 2018).

**Optimisation.** We compare models at *fixed target bitrates* rather than fixed Lagrange multipliers. We achieve this via dual ascent (Boyd et al., 2011), introducing a learnable parameter $\lambda_R$ updated iteratively:

$$\lambda_R^{(t+1)} \leftarrow \max\Big(0, \lambda_R^{(t)} + \eta_\lambda \big(R(\hat{z}) - R_{\text{target}}\big)\Big).$$

**Hyperparameter Sweeps.** We perform a grid search over the following parameters using multiple random seeds:

- **Physics Weight ($\beta$):** Swept in the range $\beta \in [0, 1]$ to trace the Pareto frontier between signal fidelity (MSE) and physical observability.

- **Target Rates ($R_{\text{target}}$):** Ranging from 0.01 to 4.0 bps with the different architecture and hyperparameter options. This validates most practical use cases (compression factors of up to $\times 1000$).

**Preprocessing and Hardware.** To normalise computational capacity and input dimensions:

- **CFD:** The native $512 \times 512$ grids are compressed, while reducing the temporal resolution to obtain less correlated samples.

- **Nyx:** We use NYX volumes of shape $512 \times 512 \times 512$ and extract 2D `xy` slices. Training/evaluation samples are $256 \times 256$ crops. Inputs are normalised with a `log1p_zscore` transform.

- **MICrONS / minnie65:** We use `xy` slices from the Minnie65 volume and extract $256 \times 256$ crops. Inputs are normalised with z-score normalisation.

Training leveraged three single-GPU workstations equipped with NVIDIA RTX 3090 (24 GB VRAM) accelerators.

## D. Additional Empirical Validation Across Domains and Observables

To broaden the empirical scope beyond 2D fluid dynamics, we evaluate the alignment diagnostic on two additional scientific domains: cosmological simulations and electron microscopy imagery. These experiments serve two purposes. First, they test whether the diagnostic remains informative across observables with very different structure. Second, they assess whether the connection between alignment and tradeoff strength persists beyond the CFD setting emphasised in the main text.

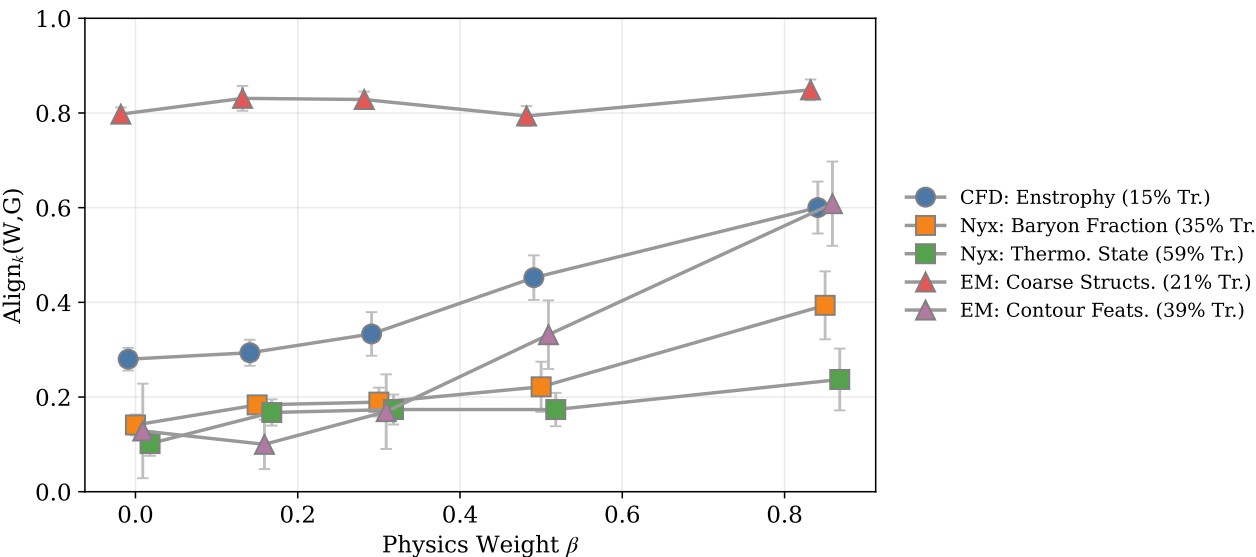

*Figure 11.* Alignment across additional domains and observables. $\text{Align}_{16}(W, G)$ evaluated across the added dataset–observable pairs under the same training sweep as in the main experiments. Percentages in the legend indicate trace coverage for the chosen rank.

### D.1. Alignment across domains and observables

We compute the alignment diagnostic $\text{Align}_k$ with fixed rank $k = 16$ using the rate-whitened metrics, and evaluate it over the same sweep of training settings used in the main experiments. Figure 11 shows that the diagnostic remains informative across all added dataset–observable pairs. As in the main text, alignment generally increases when the physics loss is more strongly weighted, but the attainable level of alignment remains highly observable-dependent. The figure shows that the diagnostic remains informative beyond CFD, but that the attainable level of alignment depends strongly on the observable: some targets become naturally compatible with signal fidelity under physics-aware training, whereas others remain weakly aligned even when the physics weight is increased.

### D.2. Alignment and tradeoff magnitude

To illustrate the associated tradeoff behaviour, Figure 12 shows the relationship between alignment and relative tradeoff magnitude for the EM observables. Each point corresponds to an operating point, with the horizontal axis reporting $\text{Align}_k(W, G)$ and marker size indicating rate. The coarse-structure observable occupies a higher-alignment, lower-tension regime, whereas the contour-feature observable lies in a lower-alignment regime with a substantially sharper tradeoff. Across both observables, increased alignment is associated with reduced tension between signal fidelity and observable fidelity.

### D.3. Summary across domains

Table 1 aggregates the main statistics across the additional domains and observables, including rate, normalised distortion trade-offs, and the alignment diagnostic. Negative values in the physics column indicate improved observable fidelity relative to the $\beta = 0$ baseline, while positive values in the MSE column indicate degraded signal fidelity. Taken together, these results support the broader claim that alignment remains predictive across scientific settings with substantially different structure, observables, and operating regimes.

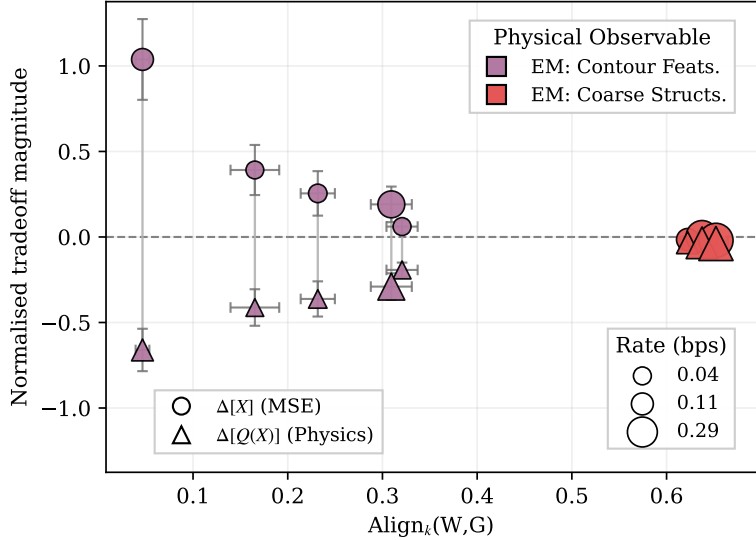

*Figure 12.* Alignment and tradeoff magnitude for EM observables. Each point is an operating point, with horizontal position given by $\text{Align}_k(W, G)$ and marker size indicating rate; circles report relative signal error and triangles report relative observable error.

*Table 1.* Cross-domain summary of the trade-off and alignment at multiple rates.

| Physical Observable | Rate (bps) | $\Delta[X]/|X_0|$ (MSE) | $\Delta[Q(X)]/|Q(X)_0|$ (Physics) | $\textbf{Align}_{16}(W, G)$ |
|---|---|---|---|---|
| CFD: Enstrophy | $0.96 \pm 0.02$ | $0.158 \pm 0.053$ | $-0.337 \pm 0.145$ | $0.275 \pm 0.079$ |
| | $0.42 \pm 0.01$ | $0.198 \pm 0.053$ | $-0.364 \pm 0.072$ | $0.488 \pm 0.129$ |
| | $0.38 \pm 0.04$ | $0.146 \pm 0.047$ | $-0.317 \pm 0.070$ | $0.518 \pm 0.096$ |
| | $0.28 \pm 0.01$ | $0.195 \pm 0.053$ | $-0.389 \pm 0.081$ | $0.545 \pm 0.160$ |
| | $0.09 \pm 0.00$ | $0.200 \pm 0.064$ | $-0.403 \pm 0.087$ | $0.492 \pm 0.173$ |
| EM: Coarse Structs. | $0.64 \pm 0.03$ | $0.014 \pm 0.024$ | $-0.048 \pm 0.031$ | $0.625 \pm 0.228$ |
| | $0.12 \pm 0.02$ | $-0.021 \pm 0.023$ | $-0.039 \pm 0.030$ | $0.624 \pm 0.111$ |
| | $0.10 \pm 0.00$ | $-0.026 \pm 0.035$ | $-0.039 \pm 0.050$ | $0.530 \pm 0.107$ |
| | $0.04 \pm 0.01$ | $-0.024 \pm 0.030$ | $-0.039 \pm 0.029$ | $0.790 \pm 0.105$ |
| EM: Contour Feats. | $0.03 \pm 0.00$ | $0.110 \pm 0.085$ | $-0.395 \pm 0.218$ | $0.138 \pm 0.276$ |
| | $0.04 \pm 0.00$ | $0.187 \pm 0.086$ | $-0.385 \pm 0.116$ | $0.165 \pm 0.325$ |
| | $0.49 \pm 0.00$ | $0.303 \pm 0.145$ | $-0.475 \pm 0.175$ | $0.032 \pm 0.044$ |
| | $0.11 \pm 0.02$ | $0.445 \pm 0.104$ | $-0.623 \pm 0.137$ | $0.048 \pm 0.092$ |
| Nyx: Baryon Fraction | $1.05 \pm 0.01$ | $-0.528 \pm 0.819$ | $-0.250 \pm 0.049$ | $0.029 \pm 0.023$ |
| | $0.15 \pm 0.01$ | $0.246 \pm 0.406$ | $-0.924 \pm 0.248$ | $0.456 \pm 0.228$ |
| Nyx: Thermo. State | $1.06 \pm 0.02$ | $-0.710 \pm 0.750$ | $0.126 \pm 0.057$ | $0.089 \pm 0.060$ |
| | $0.15 \pm 0.00$ | $-0.600 \pm 0.665$ | $-0.069 \pm 0.090$ | $0.610 \pm 0.163$ |

