# OpenReview forum: "A Geometric Lens on Physics-Aligned Data Compression"
_ICML.cc/2026/Conference — ICML 2026 regular_

### Official Review · Reviewer_ZKAM · 2026-03-07

**Soundness:** 3
**Presentation:** 3
**Significance:** 3
**Originality:** 2
**Overall Recommendation:** 3
**Confidence:** 4

**Summary:**

The paper proposes a geometric framework for physics-aware neural compression. After linearising a learned codec via Jacobians and adopting a Gaussian channel approximation, it shows that optimal bit allocation is governed by the interaction of two quadratic structures: a physics-sensitivity cost matrix $W_{\mathrm{eff}} = J^\top H J$ and a rate geometry $H_R$ from the entropy model. The main result (Theorem 4.9) gives $\Sigma^* = (H_R + \alpha W_{\mathrm{eff}} + \gamma G_{\mathrm{eff}})^{-1}$, from which the authors derive a no-free-lunch theorem (physics and signal fidelity cannot be jointly optimised when cost matrices are not spectrally aligned) and a practical alignment diagnostic ($\mathrm{Align}_k$). Predictions are validated on 2D fluid dynamics simulations from PDEBench.

**Compliance With Llm Reviewing Policy:**

Affirmed.

**Final Justification:**

I thank the authors for their rebuttal. However, after weighing the rebuttal against the submission, I am updating my evaluation for the following reasons:

**Significance and empirical scope** I acknowledge the results shared at the end of the discussion period. However, the overall experimental scope remains insufficient to demonstrate generalization of the framework beyond the original single-PDE setting.

**Originality** The rebuttal confirmed my central concern: under the paper's own linearization and Gaussian channel approximation, the local problem reduces to a classical linear-Gaussian indirect source-coding problem (Dobrushin–Tsybakov, Wolf–Ziv), and Theorem 4.9 follows from standard KKT/reverse water-filling. The authors now agree the contribution is a modelling specialization: deriving the concrete operators for neural codecs rather than new theory. This is a valid contribution, but its novelty is more incremental than the manuscript's current framing suggests.

**Soundness** The local quadratic framework is technically correct within its stated assumptions. However, the low-bitrate regime where physics-aligned compression is most practically relevant remains untested, and this is where the Gaussian approximation is most likely to break down.

In summary, the paper contains a useful practical diagnostic built on competently applied classical theory, but the combination of overclaimed novelty, limited empirical scope, and narrow validation is insufficient.

**Key Questions For Authors:**

1. **How does the paper's theoretical framework relate to the indirect rate-distortion problem of Dobrushin & Tsybakov (1962) | Wolf & Ziv (1970)?** The paper identifies the indirect R-D connection in Section 3.3 and cites Liu et al. (2021b), but does not cite or discuss the foundational works. Under the paper's own linearisation and Gaussian approximation, compressing $X$ with fidelity measured on $Q(X) \approx JX$ is an instance of indirect source coding, and $W_{\mathrm{eff}} = J^\top H J$ is the standard modified distortion from the Dobrushin-Tsybakov reduction. If there is a structural difference I am missing, or if the paper's contribution lies specifically in elements not present in the classical solution, a clear explanation would significantly help raise my assessment.

2. **Does the theory remain predictive at practical bitrates (0.1-0.5 bps)?** The hyperprior sweep begins at 1.0 bps. Empirical comparison of Theorem 4.9 predictions vs. actual codec behaviour in a lower-rate regime would address Weakness 3 and could strengthen the paper's practical contribution considerably.

3. **Can the authors demonstrate $\mathrm{Align}_k$ on at least one additional physical domain?** This diagnostic is the paper's strongest practical contribution. Showing it generalises beyond 2D fluid dynamics would substantially strengthen the empirical case and could shift my recommendation upwards.

**Limitations:**

Adequately discussed. The authors are honest about the high-rate assumption and the single experimental domain. Hyperparameter sensitivity ($\alpha, \gamma$) could be addressed more explicitly.

**Strengths And Weaknesses:**

## **Strengths**

1. **Relevant problem.** Physics-informed losses are widely used in scientific ML compression but their rate-distortion consequences are rarely analysed. The paper fills a genuine gap between the information theory and scientific ML communities, and the question of when physics-aligned training will actually help is practically important.

2. **Technically sound.** Assumptions are clearly stated, proofs are correct given their premises, and limitations are acknowledged honestly. The linearisation-Gaussian pipeline is standard and applied competently.

3. **Useful diagnostic.** The $\mathrm{Align}_k$ metric (Definition 5.3) is concrete, actionable, and convincingly validated. Computing top-$k$ eigenspace overlap via matrix-vector products without forming dense matrices (lines 358-366) is a genuine engineering contribution that practitioners can use immediately.

## **Weaknesses**

1. **Both the problem formulation and its solution have well-established classical roots.** This is the central issue. It applies at two levels:

   *Problem structure.* The paper itself identifies physics-aligned compression as an indirect rate-distortion problem (Section 3.3) and cites Liu et al. (2021b). However, the paper does not cite or discuss the foundational works that established this problem class: the indirect (remote) rate-distortion problem formalised by Dobrushin & Tsybakov (1962) [1] and solved for the additive-Gaussian case by Wolf & Ziv (1970) [2]. Under the paper's own linearisation (Section 4.1) and Gaussian approximation (Lemma 4.3), the relationship $Q(X) \approx J \cdot X$ makes this a linear-Gaussian indirect source coding problem. The cost matrix $W_{\mathrm{eff}} = J^\top H J$ is the modified distortion measure obtained by the standard Dobrushin-Tsybakov reduction from indirect to direct rate-distortion theory.

   *Solution technique.* Once reduced to a direct problem, the optimisation in Theorem 4.9 is a log-det program with linear trace constraints. The KKT derivation in Appendix A.4 (lines 636-658) is textbook: form the Lagrangian, differentiate, invert. The resulting $\Sigma^{-1} = H_R + \alpha W_{\mathrm{eff}} + \gamma G_{\mathrm{eff}}$ is a multi-constraint generalisation of reverse water-filling; the single-constraint version appears in Cover & Thomas [3] and with transform coding specifics in Goyal [4], Huang & Schultheiss [5], and Segall [6]. The extension to multiple simultaneous constraints does not introduce new structure: multi-constraint rate-distortion with trace constraints and log-det barriers is solved by the same KKT approach, as demonstrated in the semantic source coding literature (e.g., J. Liu, S. Shao, W. Zhang & H. V. Poor, "An Indirect Rate-Distortion Characterization for Semantic Sources," 2022; arXiv:2201.12477). The Bayesian sensor fusion interpretation in Appendix B.2 (Eqs. 43-44) further confirms the classical character: it recovers the Wiener filter (posterior precision = sum of prior and observation precisions).

   The paper should explicitly position Theorem 4.9 within the indirect rate-distortion literature and classical reverse water-filling, rather than presenting it as a novel geometric result. *****The genuine contribution i.e. identifying what $W_{\mathrm{eff}}$ and $G_{\mathrm{eff}}$ are for physics-aligned neural codecs, is a modelling contribution, not a theoretical one.*****

2. **Geometric language overstates novelty.** The Riemannian framing provides useful vocabulary, but the mathematics never uses genuinely Riemannian tools (parallel transport, geodesics, curvature of a manifold). $W_{\mathrm{eff}}$ is the Gauss-Newton Hessian approximation $J^\top H J$, standard in sensitivity analysis and second-order optimisation. All results operate entirely via linear algebra on tangent spaces. The paper's title and framing lead the reader to expect differential-geometric insights that do not materialise.

3. **Theory-practice gap.** The Gaussian/quadratic approximation is validated at moderate noise levels (Figure 6), but $R^2$ drops at small $\sigma$ (high rate). The regime where physics alignment matters most (aggressive compression at low bitrates) is precisely where the approximation breaks down. This is acknowledged in the limitations (page 8) but not tested below ~1.0 bps (the stated lower bound of the hyperprior sweep in Appendix C.2).

4. **Narrow experiments.** Only 2D fluid dynamics from a single benchmark. At least one additional physical domain (e.g., molecular dynamics, climate, elasticity) would test the generality of $\mathrm{Align}_k$ and the spectral alignment predictions.

5. **Isolated positioning.** The paper does not engage with the rate-distortion-perception tradeoff [7] or the rate-distortion-cognition framework [8], both of which share the multi-objective spirit of adding a third criterion beyond classical rate-distortion. More importantly, while the paper correctly identifies the indirect rate-distortion connection (Section 3.3), it does not cite or engage with the foundational works of Dobrushin & Tsybakov (1962) or Wolf & Ziv (1970), nor with the recent revival of indirect rate-distortion theory in semantic and task-oriented communication, where the problem of compressing $X$ with fidelity on $f(X)$ for linear-Gaussian sources has been solved with reverse water-filling under diagonalisability conditions (essentially the same mathematical programme as Theorem 4.9).




##### **References**

[1] R. L. Dobrushin and B. S. Tsybakov, "Information Transmission with Additional Noise," IRE Trans. Inform. Theory, 8(5),pp. 293-304, 1962.

[2] J. K. Wolf and J. Ziv, "Transmission of Noisy Information to a Noisy Receiver with Minimum Distortion," IEEE Trans. Inform. Theory, 16(4), pp. 406-411, 1970.

[3] T. M. Cover and J. A. Thomas, Elements of Information Theory, 2nd ed., Wiley, 2006.

[4] V. K. Goyal, "Theoretical Foundations of Transform Coding," IEEE Signal Processing Magazine, 18(5), pp. 9-21, 2001.

[5] J. J. Y. Huang and P. M. Schultheiss, "Block Quantization of Correlated Gaussian Random Variables," IEEE Trans. Commun. Syst., 11, pp. 289-296, 1963.

[6] A. Segall, "Bit Allocation and Encoding for Vector Sources," IEEE Trans. Inform. Theory, 22(2), pp. 162-169, 1976.

[7] Y. Blau and T. Michaeli, "Rethinking Lossy Compression: The Rate-Distortion-Perception Tradeoff," ICML, 2019

[8] J. Liu, R. Feng, Y. Qi, Q. Chen, Z. Chen, W. Zeng, X. Jin, "Rate-Distortion-Cognition Controllable Versatile Neural Image Compression," ECCV, 2024.

---

> ### Author Rebuttal · Authors · 2026-03-31
>
> We thank all reviewers for recognizing the importance of analysing rate-distortion in physics-informed ML compression. We are encouraged by the reviewers' consensus that our mathematical formulation is a “technically sound” and “useful result”, and that our Bayesian interpretation is “elegant”. Importantly, we appreciate the agreement that our proposed alignment diagnostic is a “concrete, actionable, and convincingly validated” contribution that helps practitioners “audit the AI”.
>
> Across the reviews, there was also agreement that the revision should address two main issues: clearer theoretical reframing and broader empirical scope. We agree with both. In the revision, we will (i) remove language that overstates the differential-geometric novelty and explicitly position the local derivation within classical indirect rate-distortion, and (ii) add new scientific datasets, including recently completed evaluations on Cosmological simulations and real electron microscopy of the cerebral cortex, alongside extended lower-bitrate evaluations, which confirm that our spectral alignment predictions generalize well beyond 2D fluid vorticity.
>
> ---
>
> We thank reviewer ZKAM for the careful and constructive assessment, and especially for highlighting both the strongest practical contribution and the main theoretical positioning issue.
>
> **On positioning with respect to classical indirect rate-distortion.** We agree with the reviewer’s central point. In the revision, we will explicitly reposition Section 3.3 and Theorem 4.9 within the classical indirect/remote rate-distortion literature, including Dobrushin–Tsybakov and Wolf–Ziv, and we will make clear that the novelty is not the KKT machinery itself. Our formulation already casts the problem as physics-aligned rate-distortion in which fidelity is imposed both in observable space and signal space, and Theorem 4.9 gives the resulting local optimal covariance under the paper’s linearized/Gaussian approximation. Appendix A.4 indeed derives this solution via a standard convex/KKT argument.
>
> Our intended contribution is therefore more specific: to instantiate classical indirect rate-distortion for modern learned neural codecs by deriving the concrete local operators induced by the entropy model, decoder, and physical observable—namely the local rate metric, the observable sensitivity metric, and the fidelity metric—and to show how their interaction yields both predictive tradeoff structure and a practical, matrix-free alignment diagnostic for auditing when physics-aware training should help in practice.
>
> The revision will also include positioning with adjacent modern works which share the multi-objective spirit.
>
> **On the geometric framing.** We also agree that our current language overstates the differential-geometric novelty. In the revision, we will remove the stronger “Riemannian structures” framing from the title/text and present the theory more directly as a local quadratic/operator analysis of learned compression, rather than suggesting new differential-geometric machinery. This is consistent with the actual development in Sections 4–5, which proceeds through linearization, Hessian/Jacobian pullbacks, and convex optimization.
>
>
> **On low-bitrate predictivity and broader empirical scope.** We agree that the low-bitrate regime is important and underexplored in the current submission.
>
> In the revision, we will extend the bitrate sweep down to 0.1–0.05 bps and report the agreement between the theoretical predictions and observed behavior in this regime. We will also add experiments on additional scientific domains (including recently completed evaluations on cosmology simulations and electron-microscopy cerebral cortex data) with multiple observables per dataset, and including more nonlinear/global quantities.
>
>
> **Q1. Relation to classical indirect rate-distortion:** We agree that, under our linearization and Gaussian channel approximation, the local problem is a linear-Gaussian indirect source-coding problem, and we will revise the paper to state this explicitly and cite the foundational literature directly. Our contribution is to specialize this classical structure to modern learned physics-aware codecs: we derive the concrete local operators induced by the entropy model, decoder, and physical observable; show how their interaction governs the relevant physics–fidelity tradeoffs; and operationalize this analysis into a practical alignment diagnostic.
>
> **Q2. Practical bitrates:** we agree this evidence is currently missing; we will add the results in the 0.1–0.05 bps regime and discuss more explicitly where the approximation degrades versus where the alignment diagnostic remains informative.
>
> **Q3. Additional domains:** we agree and will add new domains/observables as outlined above.

---

> > ### Author Rebuttal · Reviewer_ZKAM · 2026-04-02
> >
> > I thank the authors for the response.
> >
> > 1) I appreciate the agreement that Theorem 4.9 should be positioned within classical indirect rate-distortion literature rather than presented as a novel geometric result.
> > 2) I also welcome the plan to drop the "Riemannian structures" framing and the subjective language ("amazingly informative").
> >
> > The genuine contribution of this paper is the modelling specialization: deriving the concrete operators for neural codecs, and the practical Align_k diagnostic.
> >
> > However, a few issues remain: my main actionable concern is empirical scope.
> > It would be helpful if the authors could share the mentioned cosmology and electron microscopy results ( via anonymous link), before the end of the discussion period.
> >
> > I also agree with Reviewer hpZh's observation that Definition 5.1 conflates commutativity with the stronger condition actually needed for Theorem 5.2, which should be corrected.
> >
> > Score remains pending these clarifications.

---

> > > ### Author Response · Authors · 2026-04-08
> > >
> > > We thank reviewer ZKAM for the follow-up and for the constructive suggestions.
> > >
> > > As requested, we are happy to share the additional anonymous results here:
> > > https://anonymous.4open.science/r/icml26-results-86E5/README.md
> > >
> > > This repository contains the added cosmology and electron-microscopy experiments, including the additional observables and lower-bitrate evaluations mentioned in our response.
> > >
> > > We also agree that the current Section 5 wording is too strong: commutativity of the rate-whitened operators gives a shared eigenbasis, but does not by itself imply that the physics- and MSE-optimal allocations coincide. We will revise the definition/theorem wording to make clear that non-commutativity is only a sufficient condition for rotational conflict, while coincidence of minimizers requires a stronger spectral matching condition.
> > >
> > > We thank the reviewer again for the careful review.

---

### Official Review · Reviewer_mr3q · 2026-03-10

**Soundness:** 3
**Presentation:** 2
**Significance:** 2
**Originality:** 3
**Overall Recommendation:** 4
**Confidence:** 3

**Summary:**

This paper introduces a theoretical framework to explain the rate-distortion trade-offs inherent in training learned data compressors with physics-informed losses. The authors model this problem geometrically, demonstrating that error allocation in compression is dictated by the alignment of two Riemannian structures in the latent space: a rate geometry (driven by the entropy model) and a physics sensitivity geometry (driven by the physical observables).

**Compliance With Llm Reviewing Policy:**

Affirmed.

**Final Justification:**

acknowledge the rebuttal

**Key Questions For Authors:**

see comments above

**Limitations:**

yes

**Strengths And Weaknesses:**

pros:

This paper provides a mathematical guidance to see why that happens and how to fix the design. Here is exactly the problem it helps a practitioner solve:
- Instead of training a massive AI model for weeks only to find out the physics loss ruined the overall data quality, scientists can use the paper's "Top-k Alignment Score" diagnostic. This allows them to audit the AI and predict whether the physics-aware training will actually succeed.
- The math proves that you can only save file space (achieve rate efficiency) if the physical rule you care about is highly concentrated in a few specific directions. If a scientist wants to preserve a physical feature that is flat or spread uniformly across all the data, this paper tells them to stop, as it will be just as expensive as uniformly preserving the entire signal.
- When a team's compression model is losing standard accuracy because they added a physics rule, this framework mathematically explains exactly why it is happening.

cons:
- The theoretical model is inherently local, relying heavily on second-order approximations in the latent space. As the authors note, this model may fail to capture codec behavior under coarse quantization or when utilizing highly nonlinear decoders
- it does not offer a new optimization method to proactively enforce rate-alignment or resolve the multi-objective tension during training.
- The experiments rely exclusively on 2D velocity fields from turbulent simulations, using vorticity as the single physical observable.  Verifying these geometric phenomena across a wider variety of scientific domains is necessary to confirm the theory's broad applicability.

---

> ### Author Rebuttal · Authors · 2026-03-31
>
> We thank all reviewers for recognizing the importance of analysing rate-distortion in physics-informed ML compression. We are encouraged by the reviewers' consensus that our mathematical formulation is a “technically sound” and “useful result”, and that our Bayesian interpretation is “elegant”. Importantly, we appreciate the agreement that our proposed alignment diagnostic is a “concrete, actionable, and convincingly validated” contribution that helps practitioners “audit the AI”.
>
> Across the reviews, there was also agreement that the revision should address two main issues: clearer theoretical reframing and broader empirical scope. We agree with both. In the revision, we will (i) remove language that overstates the differential-geometric novelty and explicitly position the local derivation within classical indirect rate-distortion, and (ii) add new scientific datasets, including recently completed evaluations on Cosmological simulations and real electron microscopy of the cerebral cortex, alongside extended lower-bitrate evaluations, which confirm that our spectral alignment predictions generalize well beyond 2D fluid vorticity.
>
> ---
>
> We thank reviewer mr3q for the thoughtful assessment and for highlighting what we also view as the paper’s main practical value: the framework helps explain *when* physics-aware compression should help, *when* it will induce a tradeoff with standard fidelity, and how practitioners can audit this through the alignment diagnostic.
>
> **On the locality of the model.** We agree that the theory is local and relies on second-order approximations in latent space. In the revision, we will strengthen this point further and better separate (i) the regime where the local approximation is quantitatively accurate from (ii) the broader qualitative predictions that remain useful as heuristics outside that regime. We will also expand the empirical study to lower bitrate regimes to better quantify this theory-practice gap.
>
> **On mitigation of misalignment.** We agree that the present paper is primarily diagnostic rather than prescriptive. Its goal is to characterize when physics-aware objectives are compatible with rate-efficient compression, and when they inevitably create tension with standard fidelity, rather than to propose a new alignment-enforcing optimizer or architecture. We will make this scope clearer in the revision. We also agree that mitigation is a natural next step motivated by the theory, for example through regularization or architectural choices that encourage better alignment between the relevant operators.
>
> **On experimental breadth.** We agree that the submitted version is too narrow empirically. The current experiments focus on turbulent 2D fluid data with vorticity as the observable. To address this, we will add new physical domains, including recently completed evaluations Cosmological simulation and real electron microscopy measurements of the cerebral cortex, alongside lower-bitrate evaluations and diverse, nonlinear and global observables.

---

> > ### Author Rebuttal · Reviewer_mr3q · 2026-04-03
> >
> > Thanks for rebuttal. I have increased my score

---

### Official Review · Reviewer_hpZh · 2026-03-12

**Soundness:** 3
**Presentation:** 2
**Significance:** 2
**Originality:** 3
**Overall Recommendation:** 4
**Confidence:** 4

**Summary:**

This paper observes that in learned compression with physics-informed losses, the storage cost and two notions of reconstruction error depend on the shape of the noise ellipsoid in latent space, and uses this to characterize when physics-aware training is rate-efficient (sensitivity is concentrated in few directions) versus when it necessarily trades off against signal quality (the two objectives demand precision along different directions).

**Compliance With Llm Reviewing Policy:**

Affirmed.

**Final Justification:**

The rebuttal addressed my main question about Theorem 5.2, which justifies increasing the score.

**Key Questions For Authors:**

1. **Gap between Definition 5.1 and Theorem 5.2.** If $\widetilde{W}$ and $\widetilde{G}$ commute, the paper claims they are rate-aligned and the implication (23) does not apply. But consider $\widetilde{W} = {\rm diag}(2,1)$ and $\widetilde{G} = {\rm diag}(1,2)$: these commute, yet the minimizers $(I + \lambda_W \widetilde{W})^{-1}$ and $(I + \lambda_G \widetilde{G})^{-1}$ are distinct for any $\lambda_W, \lambda_G > 0$. So, my question is does Theorem 5.2 hold when $\widetilde{W}$ and $\widetilde{G}$ commute? Also, the proof in Appendix B.1 appears to require proportionality ($\widetilde{W} \propto \widetilde{G}$), not commutativity, to ensure the minimizers coincide. What are the authors' thoughts on this?

2. **The optimal $\Sigma^\star$ is a theoretical benchmark that may not be achievable.** The paper treats $\Sigma$ as a free variable to optimize over, but it is conceivable to me that the noise covariance is implicitly determined by the neural-network architecture and training dynamics. The paper does not discuss whether trained models actually approach the optimal allocation $\Sigma^\star$, or how far they are from it. Can the authors measure this gap for their trained models?

3. **Does the theory hold for different observables $Q$?** Vorticity is a differential operator that amplifies high frequencies, producing a specific $W_{\rm eff}$ spectrum. Have you tested observables with very different structure, e.g., integral conserved quantities or highly nonlinear $Q$? Evidence that the spectral concentration and alignment conditions vary meaningfully across observable types would address my concern about the narrow experimental scope.

**Limitations:**

Yes

**Strengths And Weaknesses:**

## Strengths

1. **Soundness.** The observation that rate, physics error, and signal error all become functionals of a single object $\Sigma$ (the latent noise covariance) under second-order expansion is a useful result. Specifically, rate becomes $\frac{1}{2}{\rm Tr}(H_R \Sigma) - \frac{1}{2}\log\det\Sigma$, while both distortions take the form ${\rm Tr}(A_i \Sigma)$ for PSD $A_i$. This reduces physics-aligned compression to a finite-dimensional convex program in $\Sigma$.

2. **Soundness.** The closed-form solution $\Sigma^\star = (H_R + \alpha W_{\rm eff} + \gamma G_{\rm eff})^{-1}$ makes explicit that the optimal error covariance suppresses variance along directions where any of the three matrices has large eigenvalues, and allows more error along directions that are simultaneously rate-cheap, physics-insensitive, and fidelity-insensitive.

3. **Originality.** The Bayesian interpretation in Appendix B.2 is elegant. Rewriting $\Sigma^\star$ as the posterior covariance of a linear-Gaussian estimation problem with prior $H_R$ and observations $H_\ell^{1/2} J_Q J_g$ and $H_D^{1/2} J_g$ gives a second, independent derivation that makes the Wiener filter structure (Eq. 44) transparent.

6. **Presentation.** Choosing vorticity $Q(v) = \partial_y v_x - \partial_x v_y$ as a physical observable is natural: it involves spatial derivatives, making it genuinely more sensitive to certain error directions than pointwise MSE.

## Weaknesses

1. **Originality.** The first main theorem (Theorem 4.9) is the KKT condition for minimizing ${\rm Tr}(A\Sigma) - \log\det\Sigma$ subject to linear constraints in $\Sigma \succ 0$. The derivative $\nabla_\Sigma(-\log\det\Sigma) = -\Sigma^{-1}$ immediately gives the closed form. The other main theorem (Theorem 5.2) follows from the observation that if $\widetilde{W}$ and $\widetilde{G}$ do not commute, then $(I + \lambda_1 \widetilde{W})^{-1} \neq (I + \lambda_2 \widetilde{G})^{-1}$ for any $\lambda_1, \lambda_2 > 0$, so the two objectives have distinct minimizers. Both results require only a few lines beyond standard convex analysis.

3. **Originality.** All experiments use one PDE system (2D compressible Navier--Stokes), one observable ($Q = \nabla \times v$), and convolutional autoencoder architectures. The paper does not provides evidence for observables with different structure, e.g., conserved quantities ($\int Q(x) dx = {\rm const}$), pointwise thresholds, or spectral densities. and without further evidence it is hard to know whether the spectral concentration and alignment conditions the theory relies on actually hold in practice beyond this single setting.

5. **Significance.** The paper explains when and why physics-aware compression faces trade-offs, but does not proposing anything to mitigate them. A natural next step from the theory would be a regularizer or architectural change that encourages $\widetilde{W}$ and $\widetilde{G}$ to share eigenvectors, reducing the conflict between the two objectives - but this is neither proposed nor tested.

6. **Presentation.** I don't understand why describing Eq. 19 as ''amazingly informative'' is appropriate for a standard KKT condition. Additionally, referring to $H_R$, $W_{\rm eff}$, $G_{\rm eff}$ as ''Riemannian structures'' is misleading - the analysis never goes beyond second-order Taylor expansions and linear algebra, and no Riemannian tools (geodesics, curvature, etc) appear to be used.

---

> ### Author Rebuttal · Authors · 2026-03-31
>
> We thank all reviewers for recognizing the importance of analysing rate-distortion in physics-informed ML compression. We are encouraged by the reviewers' consensus that our mathematical formulation is a “technically sound” and “useful result”, and that our Bayesian interpretation is “elegant”. Importantly, we appreciate the agreement that our proposed alignment diagnostic is a “concrete, actionable, and convincingly validated” contribution that helps practitioners “audit the AI”.
>
> Across the reviews, there was also agreement that the revision should address two main issues: clearer theoretical reframing and broader empirical scope. We agree with both. In the revision, we will (i) remove language that overstates the differential-geometric novelty and explicitly position the local derivation within classical indirect rate-distortion, and (ii) add new scientific datasets, including recently completed evaluations on Cosmological simulations and real electron microscopy of the cerebral cortex, alongside extended lower-bitrate evaluations, which confirm that our spectral alignment predictions generalize well beyond 2D fluid vorticity.
>
> ---
>
> We thank reviewer hpZh for the technically sharp feedback. We are encouraged that the core local reduction was found sound and the Bayesian interpretation elegant. We also agree with the main points that the paper should (i) position the optimization more modestly, (ii) broaden the empirical scope, and (iii) sharpen the alignment statement in Section 5.
>
> **On the optimization results.** We agree that the current draft overemphasizes the convex optimization itself. Under the local linearization and Gaussian channel approximation, Theorem 4.9 is indeed a standard log-det program with linear trace constraints, and the closed form follows from textbook KKT conditions. In the revision we will make clear that the contribution is **not** the generic KKT step itself, but rather the specialization of this classical local rate-distortion structure to modern learned physics-aware codecs: deriving the concrete local operators induced by the entropy model, decoder, and physical observable; showing how their interaction governs the physics--fidelity tradeoffs; and turning this into a practical implicit-operator (matrix-free) alignment diagnostic.
>
> **On experimental scope.** We agree that the submitted experiments are too narrow. In the revision, we will expand the empirical section with additional domains, including the aforementioned completed evaluations with observables with different structure (e.g., nonlinear and global quantities), and extend the bitrate sweep to lower operating points.
>
> **On mitigation of misalignment.** We agree that mitigation is a natural next step. The current paper is primarily **diagnostic**: it explains when physics-aware training helps, when it induces unavoidable tension with fidelity, and how to audit that tension in trained models. We will highlight alignment-promoting regularization / architectural design as a concrete next direction motivated by the theory.
>
> **On presentation / language.** We agree that some phrasing does not accurately reflect the intended scope and should be revised for precision. In particular, we will remove subjective wording such as “amazingly informative,” and revise the “Riemannian structures” framing to avoid suggesting differential-geometric machinery that the paper does not use.
>
> **Q1 (Definition 5.1 vs. Theorem 5.2).** Thank you for this important point. We agree that the current presentation is imprecise: **commutativity alone is not sufficient for the minimizers to coincide**. If the rate-whitened matrices $\widetilde W$ and $\widetilde G$ commute, they share eigenvectors, but the physics- and MSE-optimal allocations can still differ unless the spectra are matched strongly enough (e.g., proportional in the simplest case). Thus the current text conflates two notions: **rotational mismatch** (different eigenspaces / non-commutativity) and **spectral mismatch** (shared eigenspaces but different eigenvalue profiles). We will revise accordingly: non-commutativity will be presented only as a sufficient condition for rotational conflict, not as a full characterization of tradeoff.
>
> **Q2 (Gap from the theoretical $\Sigma^\star$).** We agree that $\Sigma^\star$ should be interpreted as a **local optimal benchmark for the surrogate problem**, not as something guaranteed to be realized by a trained neural codec. In practice, the realized covariance is shaped by architecture, the entropy-model family, finite-capacity effects, SGD dynamics, and the local-vs-global gap in the analysis. We will revise the paper to make this distinction explicit and to present $\Sigma^\star$ as a benchmark induced by the local theory rather than as an assumed training outcome.
>
> **Q3 (Different observables).**  In the revision, we include the additional evaluations as mentioned above.

---

> > ### Author Rebuttal · Reviewer_hpZh · 2026-04-04
> >
> > I thank the authors for answering all of my questions and addressing the weaknesses I raised in detail. I have revised my score now.

---

### Official Review · Reviewer_NtFx · 2026-03-12

**Soundness:** 2
**Presentation:** 2
**Significance:** 2
**Originality:** 3
**Overall Recommendation:** 4
**Confidence:** 2

**Summary:**

This paper consider geometric consideration on aligned learned compressions. They develop closed-form characterization of optimal error covariance and discuss balance physical fidelity and distortion. Also, they apply their method with turbulent fluid dynamics, showing success of simple spectral diagnostics in experimental section.

**Compliance With Llm Reviewing Policy:**

Affirmed.

**Key Questions For Authors:**

1. Is it possiblle to validate the method with real world datasets like MNIST?
2. Is it possible to compare the method with other NF methods?
3. What is the sense of corollary 4.10? could you clarify?
4. Could you give an example to clarify the sense of theorem 4.9?

**Limitations:**

high dimensional setups

**Strengths And Weaknesses:**

strengths:

1. The idea of physical alignment looks promisable


weaknesses:

1. No experiments with real world datasets like MNIST. and CIFAR-10
2. No comparison with other NF metods
3. This paper is not easy to follow, unfortunately. The sense and core of paper is a little bit hidden through the paper.

---

> ### Author Rebuttal · Authors · 2026-03-31
>
> We thank all reviewers for recognizing the importance of analysing rate-distortion in physics-informed ML compression. We are encouraged by the reviewers' consensus that our mathematical formulation is a “technically sound” and “useful result”, and that our Bayesian interpretation is “elegant”. Importantly, we appreciate the agreement that our proposed alignment diagnostic is a “concrete, actionable, and convincingly validated” contribution that helps practitioners “audit the AI”.
>
> Across the reviews, there was also agreement that the revision should address two main issues: clearer theoretical reframing and broader empirical scope. We agree with both. In the revision, we will (i) remove language that overstates the differential-geometric novelty and explicitly position the local derivation within classical indirect rate-distortion, and (ii) add new scientific datasets, including recently completed evaluations on Cosmological simulations and real electron microscopy of the cerebral cortex, alongside extended lower-bitrate evaluations, which confirm that our spectral alignment predictions generalize well beyond 2D fluid vorticity.
>
> ---
>
>
> We thank reviewer NtFx for highlighting both the promise of the physics-alignment idea and the need to make the paper easier to follow. We agree that the current presentation can be clearer, and in the revision we will clarify the framing and add an intuitive example around the main results.
>
> **On MNIST/CIFAR-10.** We agree that broader empirical validation is important. However, our target setting is **scientific / physics-aware compression**, where the key issue is preserving physically meaningful observables or quantities of interest rather than generic visual fidelity. For that reason, we believe additional scientific domains are a more appropriate testbed than MNIST/CIFAR-10.To address this, we have already evaluated our framework on additional physical datasets, including Cosmological simulation data and real electron microscopy measurements of the cerebral cortex. These new results validate our diagnostic across diverse, nonlinear observables, and will be fully detailed in the revision.
>
> **On comparisons to other methods.** We agree that the paper should be positioned more clearly relative to adjacent approaches. Our contribution is primarily **theoretical and diagnostic**: we analyze when a physics-aware objective should help, when it should induce a tradeoff with standard fidelity, and how this can be audited from the learned model. We will improve the positioning in the revision, both with respect to classical indirect rate-distortion and adjacent modern multi-objective compression perspectives, so that this distinction is clearer.
>
>
> **On the meaning of Corollary 4.10.** Corollary 4.10 makes Theorem 4.9 easier to interpret. In rate-whitened coordinates, the rate metric becomes isotropic, so the remaining structure comes from the physics and fidelity operators. This makes clear that the key issue is not only their magnitude, but also how their dominant directions align. Put differently, Corollary 4.10 identifies the right coordinates in which to compare physics sensitivity and standard fidelity, which are defined as whitened metrics $\widetilde W$ and $\widetilde G$.
>
> **An example for Theorem 4.9.** A practical way to read Theorem 4.9 is through a simple autoencoder-based compressor. An input $x$ is encoded to a latent code $z$, quantized/noised around $z$, and decoded back to $\hat x$. Around this operating point, the theorem says that the optimal local error ellipsoid in latent space is determined by three competing effects: $H_R$ (directions expensive in rate), $W_{\mathrm{eff}}$ (directions where perturbations strongly damage the physical observable), and $G_{\mathrm{eff}}$ (directions where perturbations strongly damage standard fidelity). The optimal covariance is
> $\Sigma^\star = \left(H_R + \alpha W_{\mathrm{eff}} + \gamma G_{\mathrm{eff}}\right)^{-1}$.
> So if a latent direction is cheap in rate but strongly affects vorticity, the codec should still allocate **little** variance there; conversely, if a direction is cheap and only weakly affects both vorticity and MSE, it can tolerate **more** variance. We agree that adding an intuitive example like this would improve readability, and we will include one in the revision.

---

> > ### Author Rebuttal · Reviewer_NtFx · 2026-04-01
> >
> > I am rxceedingly grateful for your answers, I understood. You totaly solved my concerns. I am intended to raise my score

---

### Decision · Program_Chairs · 2026-04-30

**Decision:**

Accept (regular)

**Comment:**

- the submission considers an important problem: understanding the rate-distortion consequences of physics-informed objectives. The main strength, identified by reviewers, is the local covariance-based analysis; also, a useful specification of this analysis to learned physics-aware codes; diagnostic for auditing when a physics-aware loss is likely to help

- at the same time, there are some concerns:
a) under the considered assumptions the main optimisation is closely related to classical indirect rate-distortion and standard convex/KKT reasoning:
b) a statement on rate alignment in Section 5 should be re-considered - commutativity alone is not sufficient for coincidence of the relevant optima;
c) the empirical validation is limited, focusing mainly on one PDE setting, and one principal observable

At the same time these concerns I consider as limitations rather than flaws that invalidates the main technical content.

- Overall, I think that the presented framework is technically sound enough to support the main message, the diagnostic contribution is sufficiently concrete and useful to present it to community.